# A unified architecture for quantum lookup tables

Shuchen Zhu

*Microsoft Quantum, Redmond, Washington, 98052, USA*
*Department of Mathematics, Duke University, Durham, NC 27708, USA and*
*Department of Computer Science, Georgetown University, Washington, DC 20057, USA*

Aarthi Sundaram and Guang Hao Low

*Microsoft Quantum, Redmond, Washington, 98052, USA*
(Dated: June 24, 2025)

Quantum access to arbitrary classical data encoded in unitary black-box oracles underlies interesting data-intensive quantum algorithms, such as machine learning or electronic structure simulation. The feasibility of these applications depends crucially on gate-efficient implementations of these oracles, which are commonly some reversible versions of the boolean circuit for a classical lookup table. We present a general parameterized architecture for quantum circuits implementing a lookup table that encompasses all prior work in realizing a continuum of optimal tradeoffs between qubits, non-Clifford gates, and error resilience, up to logarithmic factors. Our architecture assumes only local 2D connectivity, yet recovers results that previously required all-to-all connectivity, particularly, with the appropriate parameters, poly-logarithmic error scaling. We also identify novel regimes, such as simultaneous sublinear scaling in all parameters. These results enable tailoring implementations of the commonly used lookup table primitive to any given quantum device with constrained resources.

## I. INTRODUCTION

Quantum computers promise dramatic speedups over classical computers for a broad range of problems. Provable advantage in many cases such as in Hamiltonian simulation [1–6], quantum machine learning [7–12], and quantum search are in the so-called query model. The quantum gate costs of these quantum algorithms are typically dominated by queries made to a particular unitary oracle, with each oracle query having a gate cost that scales polynomially with the amount of classical data needed to encode the problem instance. As quantum computers will execute orders-of-magnitude fewer logical operations per second than classical computers [13, 14], the crossover of runtime between classical and quantum advantage is highly sensitive to the constant factors in synthesizing these oracles. Understanding the cost of oracle synthesis, such as in simulating chemistry, for a given algorithm can mean a crossover of days rather than years [15].

The dominant gate cost of quantum oracles in many cases reduces to some instance of a quantum lookup table [16] – a generic framework that facilitates access to unstructured classical data in superposition. In analogy to the classical lookup table that returns data $x_i$ for specified address bits $i \in [N]$, the quantum lookup table is some unitary quantum circuit $O_{\vec{x}}$ that responds with a superposition of data

$$O_{\vec{x}} \sum_{i\in[N]} \alpha_i \ket{i}\ket{0} = \sum_{i\in[N]} \alpha_i \ket{i, x_i}, \qquad (1)$$

when queried by an arbitrary superposition of address bits $\sum_i \alpha_i \ket{i}\ket{0}$. In general, quantum circuits implementing $O_{\vec{x}}$ can be efficiently found for any data $\vec{x}$.

It is understood that a decisive quantum advantage for interesting problems will be achieved with logical qubits in the fault-tolerant regime. Hence, the cost of quantum table-lookup should be understood in the context of fault-tolerant quantum resources. Specifically, logical Clifford gates {H, S, CNOT} are cheap, such as due to a native implementation on underlying physical qubits [17]. In contrast, logical non-Clifford gates {T, TOFFOLI} are remarkably expensive [18], following the Eastin-Knill theorem for the nonexistence of a set of transversal gates universal for quantum computing, and require sophisticated protocols to realize, like magic-state distillation and code-switching.

There are myriad possible implementations of quantum table-lookup Eq. (1), which realize distinct resource trade-offs. Early seminal work called QRAM [19] found implementations [20] using $\Theta(N)$ T gates and qubits in $\Theta(\text{polylog}N)$ depth. However, realizing this shallow depth (or runtime) is impractical due to the bottleneck of T gate production. Synthesizing a single T gate in each unit of depth has an overhead of ~100s qubits [18]. Moreover, interesting problems such as in chemistry have on the order of $N \sim 10^6$ [21] terms. Overall, the extreme space requirements make the full potential of QRAM implementations difficult to realize. This has motivated spacetime tradeoffs on the other extreme end — QROM [4] uses $\Theta(N)$ T gates, but now with the minimum of $\Theta(\log N)$ qubits in $\Theta(N)$ depth. It is straightforward to interpolate between these extremes [22], but a most surprising discovery is that interpolating a novel SELECT-SWAP [23] architecture reduces the expensive T gate count to an optimal $\Theta(\sqrt{N})$ with only a moderate number of $\Theta(\sqrt{N})$ qubits. Even more recently, the bucket-brigade QRAM implementation was discovered to be error-resilient with polylogarithmic error scaling [24] in an all-to-all quantum gate model, compared to all other approaches with linear error scaling. Error resilience indirectly reduces the cost

of logical resources, as lower errors allow cheaper, lower-distance error-correcting codes with faster logical cycle times.

Our key contribution is an architecture for table-lookup summarized in Table I that encompasses all previous methods and enables a new continuum of tradeoffs in the three key parameters of qubits, T gates, and error. Crucially, our architecture only requires a planar layout of nearest-neighbor quantum gates. There are concerns that local connectivity in QRAM variants [26, 27] limit their capabilities in end-to-end implementations [28, 29]. We similarly find in Table II, that the naive implementation of long-range gates in our architecture severely limits error resilience. However, we introduce a more sophisticated implementation based on entanglement distillation that establishes the feasibility of long-range connectivity within a planar layout. This allows us to recover, up to logarithmic factors, in Table III the scaling found in prior work assuming all-to-all connectivity. In other words, our refined method for long-range connectivity in our table-lookup architecture is equivalent to allowing unrestricted access to long-range gates, at some small (logarithmic) overhead.

Novel tradeoffs are enabled by our general architecture. These include, for instance, simultaneous sublinear scaling of all parameters with $N$, in both the case where $x_i$ is a single bit, and the more general setting of a $b$-bit word.

**Theorem I.1** (Informal version of Theorem III.1 and Corollary III.2 ). *There exists a single-word quantum lookup table that has sublinear scaling in infidelity, T-gate count, and qubit count, with local connectivity.*

**Theorem I.2** (Informal version of Theorem V.1 and Theorem V.2). *For constant word size b, there exist multi-word quantum lookup tables that have sublinear scaling in infidelity, T-gate count, and qubit count, with local connectivity.*

We also provide a fine-grained error analysis of our table-lookup implementation. We parameterize overall circuit error in terms of each common gate type present, e.g. Idling error, non-Clifford gate errors, etc. summarized in Table IV. In contrast, prior art universally assumes a generic $\epsilon$ error parameter for all gates, which we do, in Table I to ease presentation, but elaborate on with details later. Our approach proves useful for understanding dominant error sources, which future experimental implementations can then focus on minimizing.

Our table-lookup architecture is assembled from common circuit primitives reviewed in Section II, and the general framework is introduced in Section III. We then elucidate in Section IV the importance of long-range connectivity in table-lookup on a planar layout, and demonstrate that entanglement distillation provides a scalable means for overcoming the geometric error scaling of naive long-range implementations. These tools also enable our general table-lookup architecture to be extended to multi-bit words in Section V. Finally, we conclude and discuss future work in Section VI.

## II. PRELIMINARIES

All the circuit designs in this work are assumed to access data addressed by $n$ bits with a memory of size $N = 2^n$. We demonstrate how to read a single bit of data before discussing extensions for large word sizes. Here, we consider four key characteristics of the quantum lookup table and describe how they directly impact its efficiency: *circuit depth*, *qubit count*, *T-gate count*, and *infidelity*.

The minimum time taken to query a memory location is proportional to circuit depth. When referring to time, we focus on quantum operations and assume an infinite speed of light, which is a valid approximation for finite-size systems. It is commonly claimed that quantum algorithms have an exponential speedup over classical algorithms when circuit depth scales polylogarithmically in memory size. However, this minimum time is only achieved under exceptional circumstances requiring extremely large space. In most cases, the overall spacetime volume is the correct metric for evaluating cost, and not depth alone. Thus, we also consider qubit count as a key characteristic.

The qubits in a lookup table design are either used to maintain memory and router status or to act as control or ancilla bits The current generation of quantum devices is relatively small, often having a few dozen to a few hundred qubits [30, 31]. Considering such constraints in near-to-intermediate term devices, a quantum lookup table design with sublinear qubit scaling becomes highly desirable.

T-gates, essential for implementing routers in the presence of superposition queries, are non-Clifford gates that remain difficult to physically implement without substantial resources in most qubit modalities [32, 33]. Hence, in practical circuit design, the T-gate count needs to be minimized and roughly approximate the overall spacetime volume of the circuit.

The infidelity is the probability of failing a single query and it scales as a function of memory size multiplied by a generic gate error $\varepsilon$. Thus for a constant target query error, lower infidelity provides more gate error tolerance and flexibility for accommodating larger memories.

The bus bit/qubit corresponds to either the input or the output qubit storing the data. Whether it is input or output depends on the phase of the algorithm. In classical memory, a combination of logical gates sends the bus signal to the specific memory location determined by the input address bit and relays the data stored in that memory location to the output register. Similarly, quantum routers (Fig. 1) help to navigate the qubit to a memory location determined by the address, after which the corresponding classical data is loaded onto the bus qubit, and subsequently, the bus qubit is navigated to the

| Architecture | Infidelity | Query-depth | Qubits | T-gates | Layout |
|---|---|---|---|---|---|
| QRAM [19, 24] | $O(\log^2 N)$ | $O(\log N)$ | $O(N)$ | $O(N)$ | all-to-all |
| QROM [4] | $O(bN)$ | $O(N)$ | $O(\log N + b)$ | $O(N)$ | all-to-all [a] |
| SELECT-SWAP [23] | $O(bN)$ | $O(\frac{N}{\lambda} + \log \lambda)$ | $O(\log N + b\lambda)$ | $O(\frac{N}{\lambda} + b\lambda)$ | all-to-all |
| **general single-bit** (Sec. III) | $\tilde{O}(\frac{\gamma N}{\lambda})$ | $O(\frac{N}{\lambda} \log N)$ | $O(\log \frac{N}{\lambda} + \lambda)$ | $O(\frac{N}{\gamma} + \frac{N}{\lambda} \log \frac{N}{\lambda} + \lambda)$ | local, planar |
| **general multi-bit parallel** (Sec. V) | $\tilde{O}(\frac{b\gamma N}{\lambda})$ | $O(\frac{N}{\lambda}(\log bN))$ | $O(\log \frac{N}{\lambda} + b\lambda)$ | $O(\frac{N}{\lambda} \log \frac{N}{\lambda} + \frac{bN}{\gamma} + b\lambda)$ | local, planar |
| **general multi-bit sequential** (Sec. V) | $\tilde{O}(\frac{b\gamma N}{\lambda} + b^2)$ | $O(\frac{N}{\lambda} \log(bN) + b)$ | $O(\log \frac{N}{\lambda} + b\lambda)$ | $O(\frac{N}{\gamma} + \frac{N}{\lambda} \log \frac{N}{\lambda} + b\lambda)$ | local, planar |
| **single-bit** (Sec. III) | $\tilde{O}(N^{3/4})$ | $O(\sqrt{N} \log N)$ | $O(\sqrt{N})$ | $O(N^{3/4})$ | local, planar |
| **parallel multi-bit** (Sec. V) | $\tilde{O}(bN^{3/4})$ | $O(\sqrt{N} \log N)$ | $O(b\sqrt{N})$ | $O(bN^{3/4})$ | local, planar |
| **sequential multi-bit** (Sec. V) | $\tilde{O}(bN^{3/4} + b^2)$ | $O(\sqrt{N} \log(bN) + b)$ | $O(b\sqrt{N})$ | $O(N^{3/4} + b\sqrt{N})$ | local, planar |

[a] A planar layout can be found in Ref. [25] with the same resource scaling.

TABLE I: Asymptotic scaling for qubit and T gate cost, query depth, and single-query infidelity in terms of memory size $N$ and word size $b$. Trade-offs are realized by setting tuning parameters $\lambda \in [1, N]$ and $\gamma \in [1, \lambda]$ with both assuming values that are a power of 2. The types of models studied in this work are in bold. The last four rows show asymptotic results for the particular choice of $\lambda = \sqrt{N}$, and $\gamma = N^{1/4}$. The dependence on gate error $\varepsilon$ is omitted to reduce clutter. Detailed dependencies on different types of gate errors are provided in the main theorem.

| d \ d' | 0 | 1/8 | 1/4 | 3/8 | 1/2 | 5/8 | 3/4 | 7/8 | 1 |
|---|---|---|---|---|---|---|---|---|---|
| 0 | 1 | 7/8 | 3/4 | 5/8 | 1/2 | 1/2 | 1/2 | 1/2 | 1/2 |
| 1/8 | 1 | 7/8 | 3/4 | 5/8 | 9/16 | 9/16 | 9/16 | 9/16 | |
| 1/4 | 1 | 7/8 | 3/4 | 5/8 | 5/8 | 5/8 | 5/8 | | |
| 3/8 | 1 | 7/8 | 3/4 | 11/16 | 11/16 | 11/16 | | | |
| 1/2 | 1 | 7/8 | 3/4 | 3/4 | 3/4 | | | | |
| 5/8 | 1 | 7/8 | 13/16 | 13/16 | | | | | |
| 3/4 | 1 | 7/8 | 7/8 | | | | | | |
| 1/8 | 1 | 15/16 | | | | | | | |
| 1 | 1 | | | | | | | | |

| T count | 1/2 | 5/8 | 3/4 | 7/8 | 1 |
|---|---|---|---|---|---|

TABLE II: The exponent of infidelity scaling ($O(N^{(\cdot)})$) is examined for $N = 2^n$ with zero long-range budget, while varying $d$ and $d'$ subject to the constraint $d + d' \le n$, where $\lambda = 2^{n-d}$ and $\gamma = 2^{n-d-d'}$. The color gradient corresponds to different T-count scalings ($O(N^{(\cdot)})$). The $O(\sqrt{N})$ infidelity region corresponds to the regime associated with the basic planar case in Fig. 11.

| d \ d' | 0 | 1/8 | 1/4 | 3/8 | 1/2 | 5/8 | 3/4 | 7/8 | 1 |
|---|---|---|---|---|---|---|---|---|---|
| 0 | 1 | 7/8 | 3/4 | 5/8 | 1/2 | 3/8 | 1/4 | 1/8 | 0 |
| 1/8 | 1 | 7/8 | 3/4 | 5/8 | 1/2 | 3/8 | 1/4 | 1/8 | |
| 1/4 | 1 | 7/8 | 3/4 | 5/8 | 1/2 | 3/8 | 1/4 | | |
| 3/8 | 1 | 7/8 | 3/4 | 5/8 | 1/2 | 3/8 | | | |
| 1/2 | 1 | 7/8 | 3/4 | 5/8 | 1/2 | | | | |
| 5/8 | 1 | 7/8 | 3/4 | 5/8 | | | | | |
| 3/4 | 1 | 7/8 | 3/4 | | | | | | |
| 1/8 | 1 | 7/8 | | | | | | | |
| 1 | 1 | | | | | | | | |

TABLE III: The exponent of infidelity scaling ($O(N^{(\cdot)})$) is examined for $N = 2^n$ with an effectively infinite long-range budget. The linear infidelity region links to the all-to-all connectivity QRAM case explored in Ref. [4], while the zero exponent region pertains to the look-up table as detailed in Ref. [23].

output register. The circuit design of the quantum router plays a pivotal role in determining the properties of a quantum lookup table. We review some of these router designs and prior architectures each of which assumes all-to-all connectivity. We use the notation $|q\rangle_p$ to represent qubit $q$ stored in register $p$. When the context is clear, we interchangeably use $q$ and $|q\rangle_p$, or write $|q\rangle$ when the register is understood.

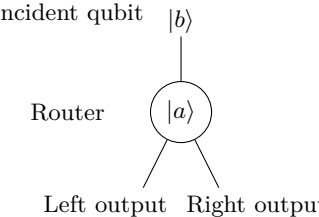

FIG. 1: A high-level description of a quantum router, the router status is set to $|a\rangle$. Incident qubit $|b\rangle$ will be routed to the left (right) when the router state is set to $|0\rangle$ ($|1\rangle$).

### A. Fan-out architecture

The fan-out architecture [34], an initial proposal for the QRAM, can be visualized as a binary tree of depth $\log N$, where the $N$ memory locations are situated at the tree's leaves (Fig. 2). Every non-leaf node in this tree functions as a router, which guides the bus signal to its left or right child. The status of the routers on the $\ell$-th level is determined by the $\ell$-th address bit, which is max-

imally entangled with the router qubits. Once the memory contents $x_j$ are written into the bus, such as using a classically-controlled X gate, the bus qubit is routed back out via the same path, and all router qubits are restored to their original disentangled state.

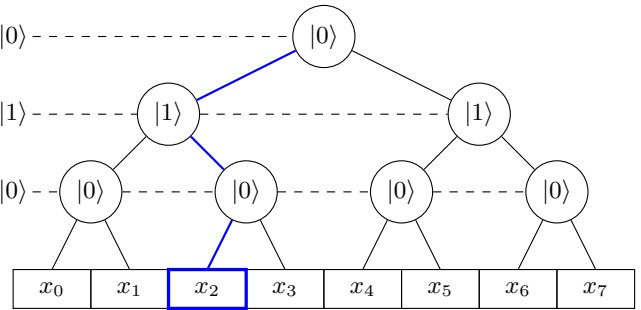

FIG. 2: High-level routing scheme for the fan-out architecture. The address bits $|010\rangle$ are entangled with their corresponding routers at each level. In the end the stored classical data at memory location $x_2$ is queried.

A significant drawback of this architecture is its high linear infidelity. If a single router gets corrupted, it will flip the status of all other routers on that same level, thereby misdirecting the query to an incorrect memory path. Consequently, for error resilience, it is suboptimal to have all the routers simultaneously entangled with the address bits.

## B. Bucket-brigade architecture

The bucket-brigade architecture [16] is an improved QRAM proposal over the fan-out architecture with three phases for querying memory: *setting router status*, *qubit route-in*, and *qubit route-out*. It uses CSWAP (Fig. 3) routers to direct signals.

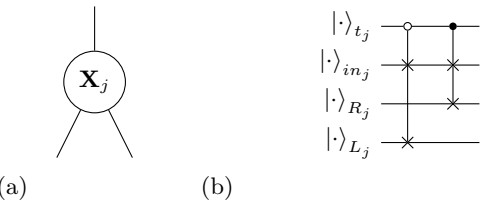

FIG. 3: (a) A single CSWAP router that takes input qubit and sends it to one of its child nodes based on its state. (b) It takes four qubits to maintain a CSWAP router. The $in_j$ register takes in input and directs the input qubit to either the left or right register depending on the state of $t_j$.

The circuit description for a CSWAP router $\mathbf{X}_j$ is illustrated in Fig. 3b following [24]. The router is composed of four qubits ($t_j$, $in_j$, $L_j$, and $R_j$) and is referred to as a CSWAP router as it uses Controlled-SWAPs to route information.

The status of a router at depth $\ell$ of the bucket-brigade model is set according to the $\ell$th address bit. For our toy model, $\mathbf{X}_j$'s status is set by storing the address qubit $a_\ell$ in the status register $t_j$. The term "activate" refers to having the routers' control qubits set to enable a path that directs any input from the top node of the binary schematic tree to one of its leaves.

During qubit route-in, the bus qubit enters the router by being swapped into the $in_j$ register. It is then swapped to the desired memory location connected to either $R_j$ or $L_j$ depending on the status in $t_j$. In Fig. 3b, the circuit for qubit route-out is not explicitly depicted, as it is the qubit-route in a circuit in reverse.

When a memory address is queried, the control qubits activate a path through the routers to the target memory cell (shown highlighted in blue in Fig. 4). Only the routers along this path are activated, significantly reducing the number of active routers at any time. This is in contrast to the fan-out architecture where all routers are activated simultaneously. As a result, only $\log N$ routers in the binary tree are strongly entangled with the address bits, and all other routers are weakly entangled. Similarly, the bus qubit storing $x_j$ is routed back out via the same path and is weakly entangled with all other routers off the path.

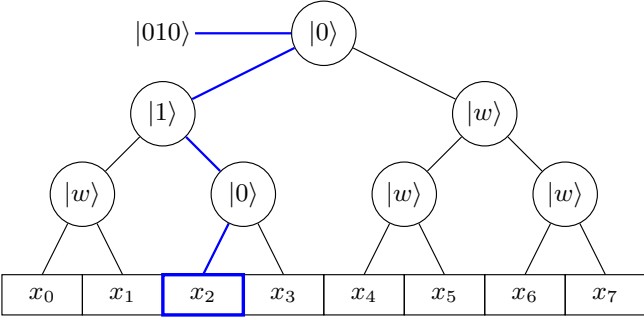

FIG. 4: High-level routing scheme for the bucket-brigade architecture. The address bits $|010\rangle$ are sent in from the root level router to set all the routers in the corresponding query path to the correct state. The routers not in the query path remain inactive.

A pivotal study by Hann et al. [24] demonstrated a circuit design for the bucket-brigade model with $O(\log^2 N)$ infidelity that is resilient to generic gate errors, assuming all-to-all connectivity. They observed that the error in some query paths remains contained and does not spread to every other branch of the bucket-brigade model thereby ensuring that some of the query paths remain free of fault. The key idea here is that CSWAP routers have a certain ability to contain error. Assuming that all CSWAP routers along a given path suffer no errors, one can prove that arbitrary errors on any other router off the path do not affect the state of the bus qubit containing the queried data. Hence, by linearity, the overall error for this model is limited only by the error of all CSWAP routers and circuit elements along any one query path.

A theoretical implementation of QRAM using superconducting circuits is presented in Ref. [35]. While preparing this manuscript, we note that a similar planar layout appears in Ref. [26, Fig. 6(c)]. However, in our work, this layout is treated as a special case within a more general architectural framework. Moreover, we provide a comprehensive theoretical analysis of this layout, including scaling behavior in qubit count, T-count, query time, and a detailed examination of infidelity at a fine-grained level.

## C. SELECT-SWAP architecture

The SELECT-SWAP architecture [23] uses a combination of linear and CNOT routers (described below) to route addresses and the fan-out architecture to route out the bus qubit.

The linear routers do not perform actual routing but instead partition the circuit into $2^d$ rounds, where $d$ is the number of linear routers. In each round $i$, they assign a value to $q_i$ in register $q$, setting it to 1 only when $i$ matches the address bits held by the linear routers as shown in Figure 5. At the start of each round $i$, register $q$ is reset to 0. The output value $q_i$ will be an activation signal propagating down the circuit. This is achieved with the use of multi-control CNOT gates and a detailed implementation can be found in Ref. [4, Section 3A]. The router is named linear as its qubits can be set in a line within a planar layout.

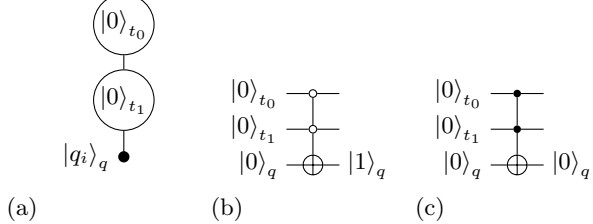

(a)      (b)      (c)

FIG. 5: (a) The linear routers $\mathbf{L}_0$ and $\mathbf{L}_1$ are configured based on the value of address bits which are set to $|00\rangle$. For two address bits, four rounds of multi-controlled CNOT gates are applied to handle all possible addresses that can occur in superposition, with $q_i$ as the target and the status of $\mathbf{L}_0$ and $\mathbf{L}_1$ as the controls. (b) For round $i = 0$, the multi-control CNOT is shown. The value of control qubit $q$ is set to 1, since $i = |00\rangle$. (c) For round $i = 3$, the value of control qubit $q$ is set to 0. This holds for any round $i \neq |00\rangle$.

The CNOT router in Fig. 6 can be used for diffusing the input signals from a parent to all child nodes. Notice that implementing a CNOT router does not require any T gates.

The high-level routing scheme for memory with 16 locations is shown in Fig. 7 where the address bits are partitioned into two sets, each controlling the linear and fan-out routers respectively. The linear and CNOT routers

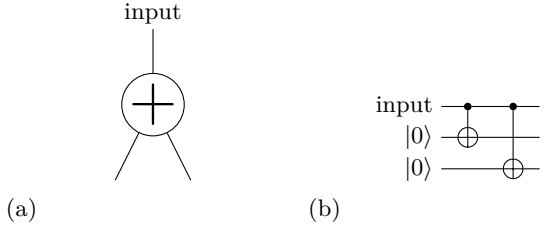

(a)          (b)

FIG. 6: (a) A single CNOT router for route-in that takes input qubit and sends it to all its child nodes. (b) Circuit for the CNOT router during route-in.

activate one out of four sets of memory locations using the first set of address bits. The second set of address bits then activates the fan-out routers to route out the qubits stored at the designated memory location to an output register.

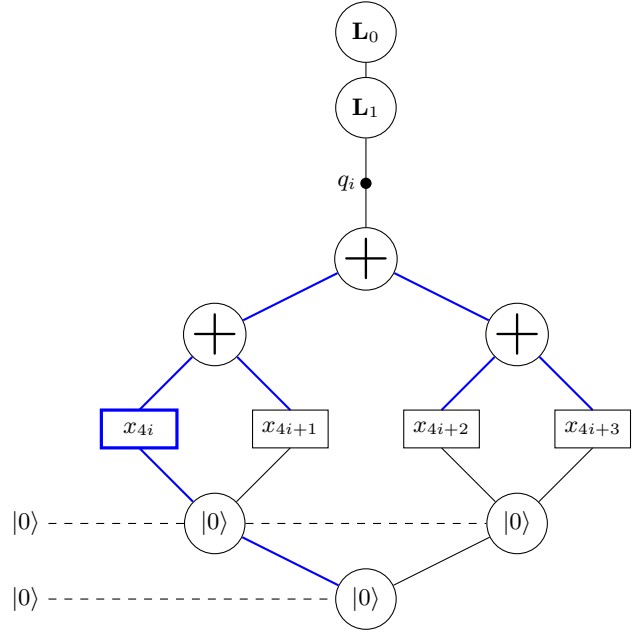

FIG. 7: High-level routing scheme for the SELECT-SWAP architecture of size $N = 16$. The detailed circuit construction can be found in [23, Fig. 1(c)]. We note that the SELECT component, implemented via linear routers, is commonly referred to as QROM [4]. Consequently, in the absence of CNOT routers, the SELECT-SWAP construction reduces to a standard QROM circuit, as shown in [23, Fig. 1(a)]. In this case, the SWAP circuit is unnecessary, since no tree structure is involved.

Although the SELECT-SWAP architecture targets T count reduction, it cannot achieve better than linear infidelity for generic error resilience. This limitation arises because every route in the tree must be correct to produce the desired output.

### D. Fine-grained error types

A common scenario while analyzing various table lookup designs is to assume a single error parameter that represents the error contribution of any operation. However, this obscures the fact that every operation could contribute differently to the errors in the overall circuit. Separating the contributions from various errors allows for more in-depth circuit profiling that helps in identifying which operations hamper the overall performance of a model or circuit design and how one can best harness its full potential. The errors that we will consider throughout this work are summarized in Table IV.

### E. Circuit optimization

We note that all the high-level circuits we have presented usually admit straightforward circuit optimizations. We outline a few examples in this section.

The CNOT router in Fig. 6a is depicted is sending a single input to two outputs. However, it can be clearly optimized to have one fewer qubit by instead having the input as one of the outputs.

Similarly, the CSWAP router Fig. 3a is depicted as performing a controlled swap to move the qubit $in_j$ to either the left or right. However, this can also also be optimized by removing one output qubit and one controlled-swap gate by having the $in_j$ input be the same as the $L_j$ output. Then, the qubit will be swapped to $R_j$ if and only if $t_j$ is 1 without affecting the functionality of the router.

| Error type | Error rate symbol |
|---|---|
| Idling | $\varepsilon_I$ |
| Qubit | $\varepsilon_Q$ |
| Long range | $\varepsilon_L$ |
| SWAP gate | $\varepsilon_s$ |
| CSWAP gate | $\varepsilon_{cs}$ |
| CNOT gate | $\varepsilon_c$ |
| CCNOT gate | $\varepsilon_{cc}$ |

TABLE IV: Error types considered in the fine-tuned analysis throughout this work. The qubit error refers to the error that occurs after a qubit has been acted on by a quantum gate. The idling error refers to the error that accumulates on a qubit during a time step when no gate is applied. The long-range error is the error on performing a long-range operation between the two qubits, as specifically defined in Corollary IV.3.

## III. GENERAL TABLE LOOKUP FRAMEWORK

In this section, we present our general error-resilient quantum table lookup architecture that can simultane-

ously have sub-linear scaling in qubit count, T count, and infidelity for a specific choice of parameters. We begin by describing our framework's structure followed by delineating its working and correctness.

The high-level scheme of our design to query a memory of size $N = 2^n$ with partition size $\lambda = 2^{n-d} \leq N$ and CNOT tree size $\gamma = 2^{n-d-d'} \leq \lambda$ can be visualized as the tree-like structure shown in Fig. 8, where $d$ and $d'$ are the depth of linear and CSWAP router, respectively. The top of our design contains $d \leq n$ linear routers $\mathbf{L}_0, \ldots, \mathbf{L}_{d-1}$ where $d = \log_2\left(\frac{N}{\lambda}\right)$. This is followed by a tree of depth $d'$ made up of CSWAP routers $\mathbf{X}_0, \ldots, \mathbf{X}_{2^{d'+1}-1}$ where $d' = \log_2\left(\frac{\lambda}{\gamma}\right)$. Each of the $2^{d'+1}$ leaves of this tree has a corresponding CNOT tree with $\gamma$ leaves attached to it. Essentially, the linear routers are used to partition the $N$ memory locations into sets of size $\lambda$ and the CSWAP routers further partition these into sets of size $\gamma$. Each leaf of the CNOT tree is connected to a memory location. The bottom of the design contains a tree of depth $n - d$ made up of CSWAP routers $\mathbf{X}_0, \ldots, \mathbf{X}_{2^{n-d+1}-1}$ to read the queried data from the appropriate location. The parts of the tree that correspond to the query path for $x_0$ are highlighted in blue in Fig. 8.

Data lookup in our framework can be broken into three stages and without loss of generality, we describe the process to query address $|a\rangle = |a_0 \ldots a_{n-1}\rangle = |0 \ldots 0\rangle$:

*a. Stage I (address setting).* The status of the linear routers is set using the first $d$ address bits such that $|\mathbf{L}_z\rangle = |a_z\rangle$. Next, the status of the $d'$ CSWAP routers in the query path are set sequentially using the address bits $|a_d \ldots a_{d+d'-1}\rangle$. Note that only the CSWAP routers along the query path are set, as opposed to every router in the CSWAP tree.

*b. Stage II (querying memory).* Let $[m]$ denote the set $\{0, \ldots, m-1\}$ for a number $m$. The objective in this stage is to compute intermediate values in registers $|\cdot\rangle_{q'_0}, |\cdot\rangle_{q'_1}, \ldots, |\cdot\rangle_{q'_{\lambda-1}}$, labeled as rectangular boxes connected to CNOT router with blue lines in Figure 8, through $N/\lambda$ repetitions. For repetition round $i \in \left[\frac{N}{\lambda}\right]$, the control qubit $q_i$ in register $|\cdot\rangle_q$ is set to $|1\rangle_q$ if and only if $i = a_0 \ldots a_{d-1}$. The control qubit $q_i$ in register $|\cdot\rangle_q$ is then routed along the query path determined by the status of the CSWAP routers set in Stage I to a leaf of the depth $d'$ tree. Then, $q_i$'s value is diffused to the $\gamma$ leaves of the activated CNOT tree attached to this leaf. In our example, $q_i$ will be routed and diffused to the leftmost CNOT tree. Note that in the $i$th repetition, the leaves of this CNOT tree are associated with the memory locations $\{\lambda \cdot i + j \mid j \in [\lambda]\}$. Finally, $q_i$'s value acts as a control indicating whether data from memory is loaded into the corresponding $|\cdot\rangle_{q'_j}$ qubit registers. For instance, when $|a_d \ldots a_{d+d'-1}\rangle = |0 \ldots 0\rangle$, qubit registers $|\cdot\rangle_{q'_j}$ are updated with the data $x_{\lambda \cdot i + j}$ for $j \in [\gamma]$. By contrast, the qubit registers $|\cdot\rangle_{q'_j}$ for $j \in \{\gamma, \gamma+1, \ldots, \lambda-1\}$ remain unchanged. Specifically, the values $q'_j$ in the $|\cdot\rangle_{q'_j}$

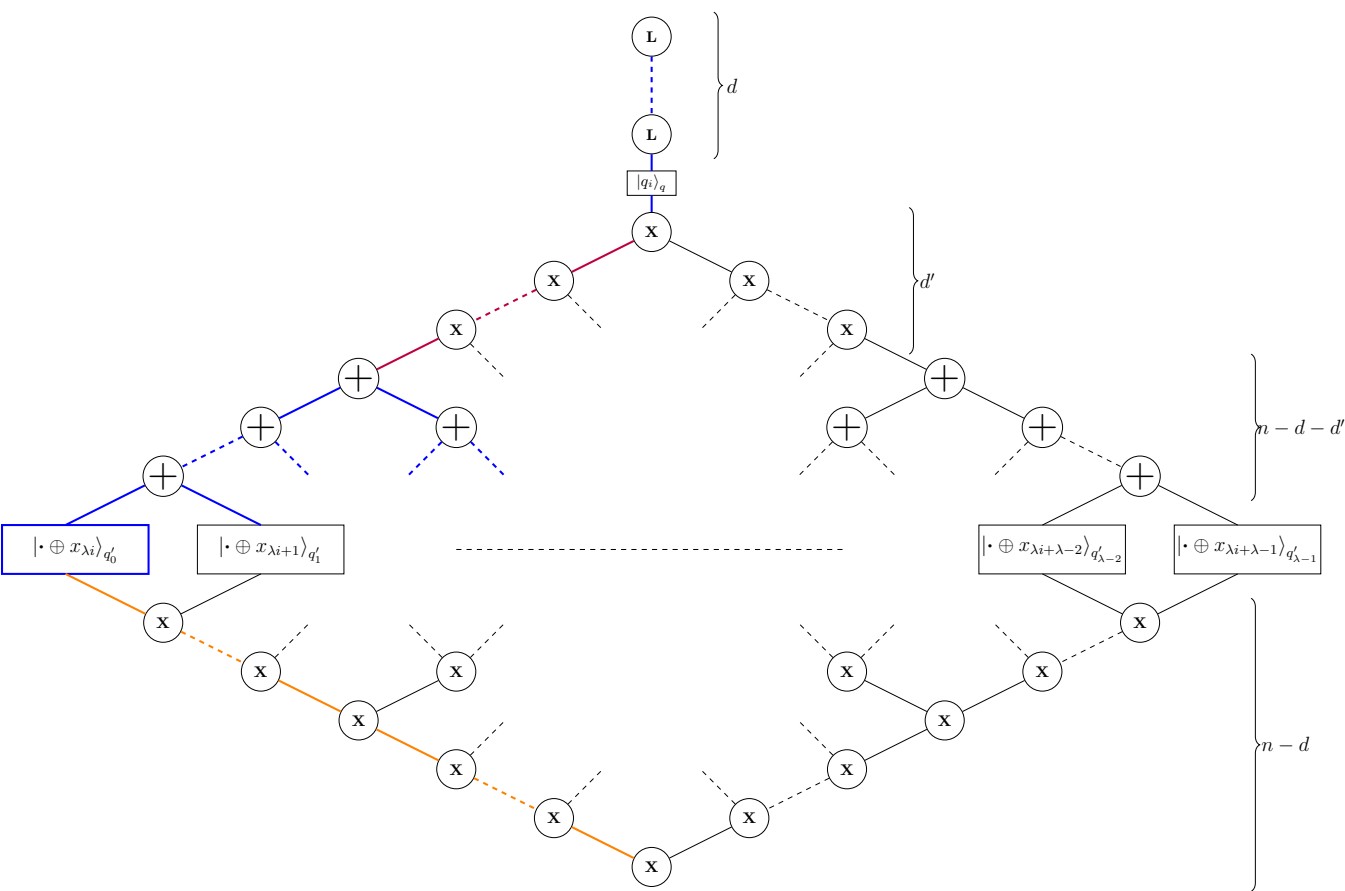

FIG. 8: High-level layout of our memory query design for $N = 2^n$ addresses, using partition size $\lambda = 2^{n-d}$ and CNOT tree size $\gamma = 2^{n-d-d'}$. The top layer consists of $d$ linear routers that partition memory into blocks of size $\lambda$. This is followed by a depth-$d'$ CSWAP tree, where each leaf connects to a CNOT tree with $\gamma$ leaves. A final CSWAP tree of depth $n - d$ routes the queried data from memory to the output. The query algorithm consists of three stages, with details provided in the text of Section III. The corresponding algorithm is shown in Algorithm 1. The purple line is used to indicate path $p_1$ and the orange line is used to indicate path $p_2$ in Algorithm 1.

registers satisfy:

$$q_j' = \bigoplus_{i=0}^{N/\lambda} q_i x_{\lambda \cdot i + j} \qquad (2)$$

where $\oplus$ denotes addition modulo 2. For our example, after $N/\lambda$ repetitions, only $q_0 = 1$ and only the leftmost CNOT tree is activated. Hence, we find that $|q_j'\rangle_{q_j'} = |x_j\rangle_{q_j'}$ for $j \in [\gamma]$ and $|q_j'\rangle_{q_j'} = |0\rangle_{q_j'}$ otherwise. We remark that between each repetition, the qubits in the CNOT trees can be reset to $|0\rangle$ using measurement-based uncomputation. Additionally, $q_i$ in its corresponding register $|\cdot\rangle_{q_j'}$ is routed back to the top of the CSWAP tree and all qubits except for the router status qubits are reset to $|0\rangle$.

*c. Stage III (retrieving data).* Data is retrieved similar to how the bus qubit is routed out of the noise-resilient bucket brigade architecture in Ref. [24]. We use the address bits $|a_d \ldots a_{n-1}\rangle$ to set the status of the $n - d$ CSWAP routers in the entire tree, specifically, some of the CNOT routers (of total depth $n - d - d'$) in Stage II are reconfigured as CSWAP routers. As in Stage I, it is only the routers in the query path whose status is set. The leaves of this CSWAP tree point to the $|\cdot\rangle_{q'}$ registers and only the data in $|\cdot\rangle_{q_\ell'}$ for $\ell = a_d \ldots a_{n-1}$ is retrieved. In our example, this leads to data in register $|\cdot\rangle_{q_0'}$ being retrieved at the end of the data lookup process.

Note that repetition is confined to Stage II, where the registers $|\cdot\rangle_{q_j'}$ corresponding to the partial address bits are loaded with the appropriate lookup table data, facilitated by the CNOT tree. In Stage III, the remaining address bits configure the CSWAP routers to output the classical data stored in register $|\cdot\rangle_{q_j'}$ corresponding to the full address bits, and this process occurs only once. When $d' = n$, it reduces to QRAM; for $d = n$, it reduces to QROM; and for $d' = 0, d < n$, it reduces to SELECT-SWAP.

The detailed procedure for data lookup is shown in Algorithm 1. Our objective is to examine the optimal

balance between $d$ and $d'$ that results in the most favorable infidelity, T count, and qubit count scaling. Our fine-grained analysis uses the error types from Table IV.

**Theorem III.1.** *Consider the quantum data lookup structure with the high-level scheme in Fig. 8 with $N$ memory locations. Let $n = \log N$, $\lambda = 2^{n-d}$ be the partition size and $\gamma = 2^{n-d-d'}$ be the size of a CNOT tree with $d' \leq d \leq n$. The infidelity of this circuit is*

$$
O\left(\varepsilon_L \left(\frac{\gamma N}{\lambda} + \frac{N}{\lambda} \log \frac{\lambda}{\gamma}\right) + \varepsilon_s \log \frac{\lambda^2}{\gamma}\right.
$$
$$
+\varepsilon_I \left(\frac{N}{\lambda} \log N \left(\log \frac{N}{\gamma} + \gamma + \log \frac{\lambda}{\gamma}\right) + \text{polylog } \lambda\right)
$$
$$
+\varepsilon_c \frac{\gamma N}{\lambda} + \varepsilon_{cc} \frac{N}{\lambda} \log \frac{N}{\lambda}
$$
$$
\left.+\varepsilon_{cs} \left(\frac{N}{\lambda} \log \frac{\lambda}{\gamma} + \log^2 \frac{\lambda}{\gamma} + \log^2 \lambda\right)\right). \tag{3}
$$

*Moreover, the T count for this design is $O(\frac{N}{\gamma} + \frac{N}{\lambda} \log \frac{N}{\lambda} + \lambda)$, and its qubit count is $O(\log \frac{N}{\lambda} + \lambda)$.*

We will restate and prove this theorem in Section IV C after explaining how to lay out the scheme in Fig. 8 on a planar grid with nearest neighbor connectivity. Here, we use the above theorem to find an instance that has sub-linear scaling for infidelity, qubit, and T-counts.

**Corollary III.2.** *For $N$ memory locations, there exists a quantum data lookup scheme that has $\tilde{O}(N^{\frac{3}{4}})$ infidelity, $O(N^{\frac{3}{4}})$ T count, and $O(\sqrt{N})$ qubit count.*

*Proof.* For the high-level scheme depicted in Fig. 8, setting $\lambda = \sqrt{N}$ and $\gamma = N^{1/4}$ and applying Theorem III.1, gives the result. $\square$

We claim that it is necessary to have CSWAP routers at the top of our design in Stages I and II to achieve sublinear infidelity scaling. Assume by way of contradiction that the CSWAP routers are replaced with CNOT routers. First, note that CNOT routers are not robust to Pauli $Z$ errors as shown in Fig. 9. Although the Pauli $Z$ error propagates only to the parent node in the CNOT router, it can result in a phase kickback that can alter the address state presented during a query. Specifically for our example, this happens when $i = a_0 \ldots a_{d-1}$ in Stage II, and there are an odd number of Z errors along the query path in the framework. A comparable scenario is also presented in Ref. [26, Section V], where the readout CNOT tree demonstrates resilience to only Pauli Z errors.

Second, to prevent a phase kickback, we need to assume that the entire CNOT tree with $\lambda$ leaves is part of the query branch and remains Pauli $Z$ error-free. In this case, the idling error will be dominated by Stage II's contribution of $O\left(2^d \varepsilon_I \left(d + 2^{n-d}\right)\right) = O(2^n)$ leading to a linear infidelity scaling. By contrast, the CSWAP router is robust against error propagation (see Section IV for details). Hence, we consider our framework with a non-zero number of CSWAP routers at the top of our design to be a more resource-efficient approach.

Uncomputing the table lookup circuit is crucial to ensure that there are no residual garbage states entangled with the address and output registers once a query has been performed. For our design, uncomputing can be done as follows. First, run the circuit for Stage III in reverse to route the output bit back into its corresponding $q'$ register, then run the Stage II circuits again to set all $q'$ registers to 0. Finally, run the Stage I circuit in reverse to reset the status of all the routers. This will effectively double the infidelity scaling.

A potential way to reduce the T count for our framework without worsening its query infidelity is by modifying the design for Stage III. Specifically, in Fig. 8, we retain the CSWAP tree from the bottom of the figure up to a depth $d'$. Each of the $2^{d'}$ leaves of this tree has a corresponding tree of fan-out routers with $\gamma$ leaves attached to it. Each of the $\gamma$ leaves is connected to a corresponding $q'_j$ register. Essentially, this creates $\lambda/\gamma$ different fan-out router substructures each of whose routers is set independently of the other. Let $\ell = a_d \ldots a_n - 1$. Then, this will impose the condition that the $\gamma$-sized sub-structure of the fan-out routers containing $q'_\ell$ should be error-free for noise resiliency. Since a similar condition is satisfied by the CNOT trees in Stage II, this does not affect the asymptotic scaling for the infidelity. However, the T count reduces as the fan-out routers do not use T gates. A more detailed analysis of this improvement is left for future work.

## IV. PLANAR LAYOUTS FOR QUANTUM DATA LOOKUP FRAMEWORKS

In this section, we discuss how our general quantum data lookup framework can be designed on a planar layout with only local connectivity. We first build some intuition for the underlying principles to achieve this by modifying the bucket-brigade design of [24] for a planar layout. This is illustrated in Section IV A. An initial analysis of query infidelity for this design, shows that it scales sub-linearly in memory size for the planar layout. In Section IV B we use entanglement distillation to perform long-range operations and recover the log scaling for query infidelity in the planar layout. We put these ideas together to present the planar layout for our general framework in Section IV C. In the figures illustrating the planar layout, blue lines are used to represent connections between all registers within a single router, while red lines indicate connections between routers at different levels.

---

**Algorithm 1** Pseudocode for quantum data lookup on memory of size $N$, partition size $\lambda$, and CNOT tree size $\gamma$. This corresponds to the high-level routing scheme shown in Figure 8. SETROUTER($a$, $r$) sets the address bit(s) $a$ to router(s) of type $r \in \{\mathbf{X}, \mathbf{L}\}$ as described in Section II. For each SETROUTER operation, a path connected by its corresponding routers is formed. ROUTEDATA($d$, $p$) moves data qubit $d$ from one end to the other end of the path $p$. We use $\tilde{p}$ to denote the reverse of the path $p$.

---

**Setup:** classical database data of size $N$ with $n = \log_2 N$, memory partition $\lambda \leq N$ with $d = \log_2\left(\frac{N}{\lambda}\right)$ and CNOT tree size $\gamma \leq \lambda$ with $d' = \log_2\left(\frac{\lambda}{\gamma}\right)$.
**Input**: state $|a\rangle_{\text{add}}$ where $a = a_0 \ldots a_{n-1}$ is an address of length $n$.
**Output**: state $|a\rangle_{\text{add}} |\text{data}[a]\rangle_{\text{out}}$ where data$[a]$ refers to the data at address $a$.

    **Stage I:**
1: Initialize all registers of the quantum lookup table to $|0\rangle$.
2: SETROUTER($a_0 \ldots a_{d-1}, \mathbf{L}$)                                         ▷ Set $d$ linear routers $\mathbf{L}_k$
3: SETROUTER($a_d \ldots a_{d+d'-1}, \mathbf{X}$)           ▷ Set $d'$ CSWAP routers $\mathbf{X}$ in the corresponding path $p_1$ of the CSWAP tree
    **Stage II:**
4: **for** $i$ in $0 \ldots 2^d - 1$ **do**
5:     **if** $i = a_0 \ldots a_{d-1}$ **then**             ▷ The state of Linear routers $\mathbf{L}$ determines the activation qubit $q_i$ at round $i$
6:          Set $|q_i\rangle_q \leftarrow |1\rangle$
7:     **else**
8:          Set $|q_i\rangle_q \leftarrow |0\rangle$
9:     ROUTEDATA($q_i$, $p_1$)              ▷ Route $q_i$ along the path $p_1$ determined by the CSWAP routers to a CNOT router
10:     Propagate $q_i$'s signal to the $\gamma$ leaves $|c_0, \ldots, c_{\gamma-1}\rangle_{q'}$ of the CNOT tree.
11:     **for** $j$ in $0 \ldots \lambda - 1$ **do**                               ▷ Set the $q'$ registers determined by $q_i$ and data
12:          Apply unitary $U_{\lambda \cdot i+j}$ where $U_{\lambda \cdot i+j} = \text{CNOT}|c_j\rangle |q'_j\rangle$ if data$[\lambda \cdot i + j] = 1$ and $\mathbf{I}$ otherwise.
13:     Reset all qubits $c_j$ and those in the CNOT tree to 0.
14:     ROUTEDATA($q_i, \tilde{p}_1$)                          ▷ Route $q_i$ back up the CSWAP routers to its original location
15:     Reset all the qubits of the CSWAP routers except the router status qubits and $q_i$ to 0.
    **Stage III:**
16: SETROUTER($a_d \ldots a_{n-1}, \mathbf{X}$)     ▷ Set the status of $\log_2(\lambda)$ CSWAP in path $p_2$ routers based on the remaining address bits
17: ROUTEDATA($q'_\ell, p_2$), $\ell = a_d \ldots a_{n-1}$ ▷ Route $q'_\ell$ along the query path $p_2$ set by the CSWAP routers to the output register

---

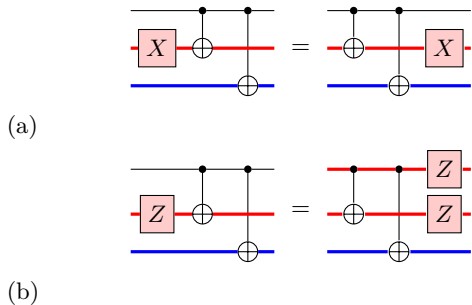

(a)

(b)

FIG. 9: (a) Pauli $X$ error does not propagate from the bad branch (red) to the good branch (blue) in the CNOT router. (b) Pauli $Z$ error does propagate into the parent register from a child branch in the CNOT router.

### A. Planar layout for the bucket-brigade model

We provide a circuit design for the bucket-brigade QRAM model, assuming the qubits are laid out on a 2D planar lattice and multi-qubit gates act only on adjacent qubits. We first demonstrate a toy model with two memory locations, where the routing scheme is depicted as a binary tree in Fig. 3a. To reach the desired memory allocation, a single CSWAP router $\mathbf{X}_0$ is employed to direct the incoming address qubit into the designated memory location, from where the stored data $x_i$ is retrieved and then routed out using the same path.

The CSWAP router's four qubits can be arranged in a *T-shaped* configuration as shown in Fig. 10a where the qubits are located at the intersections of the grid. This configuration ensures that each router qubit is adjacent to any other router qubit with at most one local SWAP. The three-qubit CSWAP gate can be decomposed into a sequence of Clifford and T gates that operate on at most two qubits [23].

For the larger memory size of $N = 16$, the high-level bucket-brigade routing scheme is shown in Fig. 11a, where both the route-in and route-out phases follow the same path in the tree. The corresponding planar layout is shown in Fig. 11b where the blue lines show the structure of a CSWAP router, the red lines show connections between different levels of the routing scheme, and the input, address, and bus qubits are positioned in the center of the diagram.

The routers are placed on the grid following the H-tree fractal pattern [36] starting from the root at the center and leaves at the boundaries of the grid. The left and

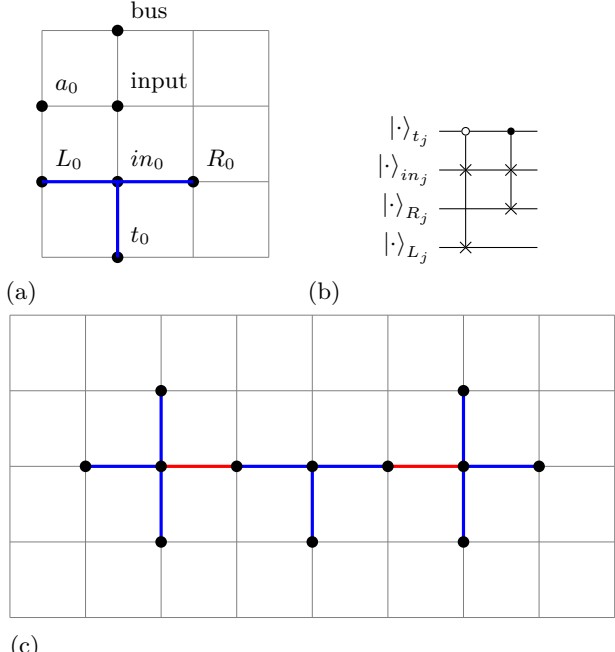

FIG. 10: (a) A planar layout of qubits for the toy model with nearest-neighbor connectivity. (b) The CSWAP circuit corresponding to the blue T-shaped layout in (a). (c) Joining two T-shaped routers to form a single H-tree segment.

right registers of the leaf-level routers send an incoming qubit to the respective memory locations. A pair of T-shaped routers each laid out according to Fig. 10a can be joined together to form a single H-tree segment as shown in Fig. 10c. Note that such a planar layout scheme can be naturally extended to higher dimensions such as a cubic grid for 3D. However, we focus only on the planar grid throughout this work.

The recursive expansion of the fractal layout yields an optimal layout that occupies an area of size $O(N)$. For this layout scheme the T count, and qubit count scale as $O(N)$ since the layout uses $O(N)$ CSWAP routers and there are $O(N)$ points in the rectangular grid where each point corresponds to a qubit. Unlike in the all-to-all connectivity case, the error accumulation in the planar layout occurs due to the long-range gates that need to be performed such as those along the red lines in Fig. 11b. Naively, if we assume that the probability of gate error is $\varepsilon_{max}$ and a long-range SWAP is performed using a successive series of SWAP gates, the overall infidelity scales as $O(N\varepsilon_{max})$ as both the circuit depth and number of gates for each query scales as $O(\sqrt{N})$. However, by performing a more fine-grained error analysis, we show how the infidelity can scale sub-linearly in $N$.

The foundation for calculating the infidelity scaling lies in the crucial property of error containment exhibited by the bucket-brigade QRAM. Consider the tree branches in Fig. 11a. They can be categorized either as good or

bad based on the presence or absence of errors in them. In Ref. [24, Appendix D], it was shown that the errors do not spread from a bad branch to a good branch, and assuming that the query path is a good branch, this implies that errors in other parts of the QRAM do not significantly affect the query. Importantly, this holds regardless of the layout scheme as long as CSWAP routers are used to enact the high-level routing scheme in Fig. 11a. Hence, we can use the error containment property for our planar layout too.

To improve the overall infidelity of our layout scheme, we modify the circuit and employ constant depth circuits for long-range operations between non-adjacent qubits. One way to implement a long-range SWAP between two qubits separated by a line of $m$ qubits(e.g., a single red line in Fig. 11b), is to use a strongly entangled length-$m$ GHZ state as a resource. The long-range gate is then performed involving the qubits near the endpoint as shown in Ref. [37]. A length-$m$ GHZ state is

$$|\text{GHZ}_m\rangle = \frac{|0\rangle^{\otimes m} + |1\rangle^{\otimes m}}{\sqrt{2}}. \qquad (4)$$

The error contribution for using GHZ states in this case is stated below.

**Lemma IV.1.** *For a given GHZ state of length $m$, the probability that any long-range operation using this state has an error is $O(m\varepsilon_Q)$ where $\varepsilon_Q$ is the probability of a single qubit having an error.*

*Proof.* For the GHZ state to be correct, all its underlying qubits have to be correct. Using triangular inequality yields the desired error probability for the GHZ state. $\quad\square$

Lemma IV.1 shows that despite the constant depth needed for the long-range operation, its error rate increases linearly with the length of the GHZ state. However, GHZ states of arbitrary length can be created using a constant-depth circuit [38, Sec. 5.1]. Consequently, all the GHZ states utilized for the long-range SWAPs can be generated in place without incurring a substantial overhead. In fact, with this modification, the circuit depth for the planar layout reduces from $O(\sqrt{N})$ to $O(\log N)$, thereby leading to a sub-linear infidelity scaling of $O(\sqrt{N})$.

## B. Recovering log scaling in infidelity

As using GHZ states still gives a polynomial dependence on $N$ in the infidelity analysis, we instead consider performing a long-range SWAP on remote qubits using a Bell state $\left(\text{i.e., } |\Phi^+\rangle = \frac{|00\rangle + |11\rangle}{\sqrt{2}}\right)$ between them as a resource. To obtain a high-quality Bell state, we use noisy Bell states, a quantum error correcting code, and an entanglement distillation protocol as shown in [39, Section II.D].

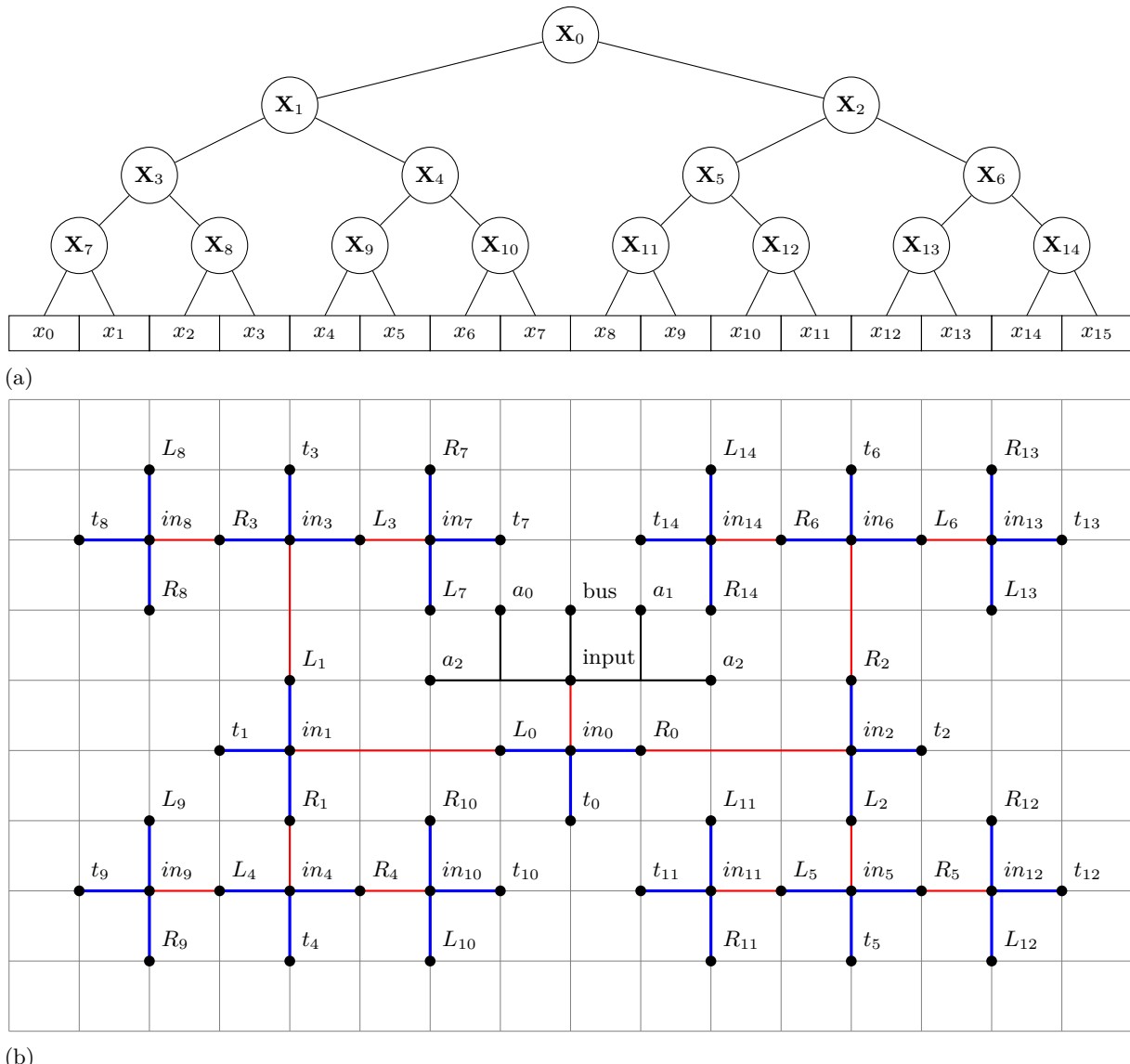

(a)

(b)

FIG. 11: (a) High-level routing scheme for the bucket-brigade architecture with 16 memory locations. (b) Qubit layout arrangement of the bucket-brigade model with 16 memory locations.

**Lemma IV.2.** *Given an* $[[\hat{n}, \hat{k}, \hat{d}]]$ *quantum error correcting code and* $\hat{n}$ *noisy Bell pairs with initial error* $\varepsilon_i$, *there exists a distillation protocol that creates* $\hat{k}$ *Bell pairs with error* $\varepsilon_f < \varepsilon_i$ *where* $\varepsilon_f = O(\varepsilon_I^d)$. *Moreover, when* $\varepsilon_i = O(m \cdot \varepsilon_G)$ *and* $\varepsilon_f < 1$ *is a small constant,* $d = O(\log m)$.

To perform a long-range SWAP, consider the Bell pair being created on adjacent qubits near the source qubit with one half of it being teleported to be adjacent to the target qubit using the length-$m$ GHZ state where $m = O(\sqrt{N})$ as per our layout. Then, from Lemma IV.1 these noisy Bell pairs could have an error $\varepsilon_i \leq O(\sqrt{N} \cdot \varepsilon_Q)$ and it is possible to distill a Bell pair with constant error using say, the surface code, with distance $O(\log N)$. For this choice of code, the protocol to distill would have

depth $O(d) = O(\log N)$ and the number of noisy Bell pairs used would be $\hat{n} = O(d^2) = O(\log^2 N)$. Combining the two methods gives the following.

**Corollary IV.3.** *Given two qubits separated by* $m$ *qubits on a planar grid with local connectivity, qubit error* $\varepsilon_Q$ *and the error on a distilled Bell pair* $\varepsilon_f$, *the error on performing a long-range operation between the two qubits is given by*

$$\varepsilon_L := \min(m \cdot \varepsilon_Q, \varepsilon_f). \tag{5}$$

We acknowledge that, by using entanglement distillation, the overall circuit depth may increase by a polylogarithmic factor in the worst case, which is acceptable. However, there might be strategies to mitigate this depth increase. Therefore, for the purposes of our analysis, we

proceed under the assumption that long-range Bell states are readily available.

Accounting for the overheads due to entanglement distillation, for the planar layout, we claim that the circuit depth $D$ scales as $O(\text{polylog } N)$. Understanding the activation sequence of routers in a query branch is beneficial in performing fine-grained error analysis. For instance, in the *setting router status* phase, we try to route as many address qubits as possible in parallel. This means that it is not necessary to wait for the address qubit $|a_\ell\rangle$ to reach the router at level $\ell$ before sending the address qubits $|a_{\ell+1}\rangle$ to be routed by $\mathbf{X}_0$. Specifically, this reveals that not all qubits need to maintain their state throughout the entire query depth $T$. Suppose we are given a fixed query branch of depth four, with the routers $\mathbf{X}_0, \mathbf{X}_1, \mathbf{X}_2, \mathbf{X}_3$ counting from root to leaf. The activation sequence for each router and associated gates is depicted in Fig. 12, where the time $\tau_i$ increases by an $O(\text{polylog } N)$ additive factor and hence the total query time $\sum_i \tau_i = O(\text{polylog}(N))$.

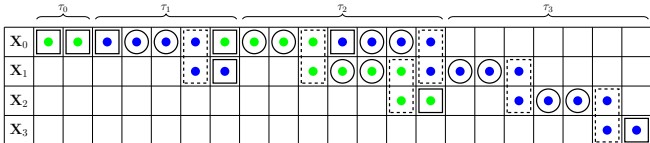

FIG. 12: The activation sequence of routers in a given query branch of depth four, where each column represents a single time step. $\tau_i$ is the total time to set the status bit for router $\mathbf{X}_i$. The green and blue dots indicate the router at the associated time step that is in use. The rectangle represents a local SWAP operation over the qubits associated with the router, the circle represents a local CSWAP operation, and the dashed rectangle represents a long-range SWAP between qubits associated with parent and child routers. The green and blue colors refer to routers at even and odd depths respectively of the tree as per the routing scheme in Fig. 11a. The circuit reference can be found in Ref. [24, Fig. 10]. The idea of routing multiple qubits into the tree simultaneously at different layers can also be found in Refs. [26, 28, 40].

The error terms for our fine-grained analysis are taken from Table IV.

**Theorem IV.4.** *For $N$ memory locations, the improved fine-grained infidelity of the bucket-brigade QRAM with planar layout (Fig. 11b) scales as*

$$O\left(\log^2 N \varepsilon_L + \log N \varepsilon_s + \log^2 N \varepsilon_{cs} + \text{polylog } N \varepsilon_I\right). \tag{6}$$

*Proof.* For a fixed query branch, we first consider the error contribution from the long-range operation. Let $T = \log N$ be the tree depth of the routing scheme as shown in Fig. 11a. By Corollary IV.3, the contribution of long-range error to the probability of a successful query

is

$$P_L = \prod_{\ell=1}^{\log N} (1 - \varepsilon_L)^{3(T-\ell)} = (1 - \varepsilon_L)^{O(T^2)}, \tag{7}$$

where $3(T-\ell)$ is the number of times a long-range CNOT is applied to execute $T - \ell$ long-range SWAP operations. This pattern is also evident in Fig. 12, where a long-range operation between $\mathbf{X}_1$ and $\mathbf{X}_2$ occurs $T$ times, but occurs only $T - 1$ times between $\mathbf{X}_2$ and $\mathbf{X}_3$.

Next, consider the error contribution from the local SWAP operation over qubits associated with individual routers. It can be observed from Fig. 12 that it takes two local swap operations for the address setting of each router. Hence the contribution of local SWAP error to the probability of a successful query is

$$P_s = (1 - \varepsilon_s)^{2T} = (1 - \varepsilon_s)^{O(T)}. \tag{8}$$

The error contribution of the local CSWAP operation can be found similarly, and its probability contribution toward success is

$$P_{cs} = \prod_{\ell=1}^{\log N} (1 - \varepsilon_{cs})^{2\ell} = (1 - \varepsilon_{cs})^{O(T^2)}. \tag{9}$$

Last, we consider the *idling error* of the status qubit $t_\ell$ for each router $\mathbf{X}_\ell$ as it must maintain its value immediately after its associated router's address is set. By contrast, the other qubits in the router can be reset and remain irrelevant until their next usage. The idling time for each $\mathbf{X}_\ell$'s status qubit $t_\ell$ is the total active time of the router minus the number of CSWAP operations over the qubits of $\mathbf{X}_\ell$. Then, by Fig. 12 and Corollary IV.3, the total idling time for $\mathbf{X}_\ell$ is $O(T - \ell + \text{polylog}(\frac{\sqrt{N}}{2^{\lceil \ell/2 \rceil}}))$. Therefore, the idling qubit error contribution toward total query success probability is

$$P_I = (1 - \varepsilon_I)^{O(\text{polylog } N)}. \tag{10}$$

Since the total success probability is $P = P_L \cdot P_s \cdot P_{cs} \cdot P_I$, combining Corollary IV.3 with a similar analysis as in reference [24, Appendix D], we obtain the desired infidelity. $\square$

### C. Planar layout for the general framework

The CSWAP routers form only one part of our general data lookup framework but the techniques from Section IV A can be reused here. For ease of description, consider a memory of size $N = 16$, partition size $\lambda = 4$, and CNOT tree size $\gamma = 2$ for our design as shown in Fig. 13.

The planar layout for 16 memory locations is given in Figs. 14a and 14b where the former depicts the layout during Stages I and II while the latter holds for Stage III. Some of the routers are labeled in the figures with their components surrounded by dashed boxes. As the

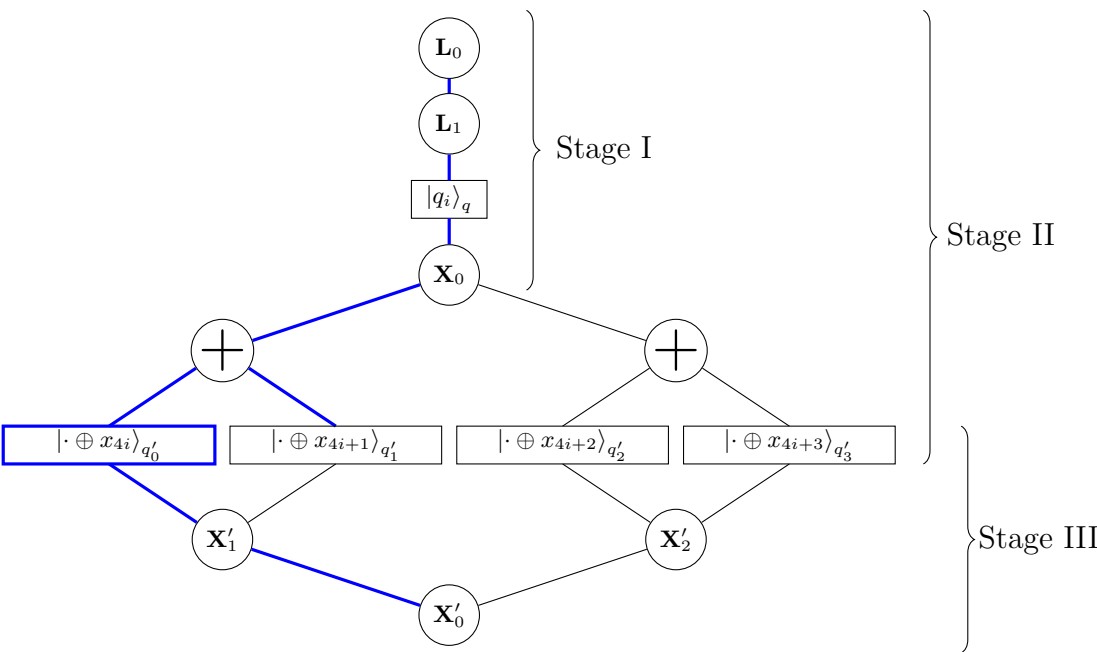

FIG. 13: High-level routing scheme for a general noise resilient data lookup framework with sub-linear scaling in qubit count, T count and infidelity having memory size $N = 16$, partition size $\lambda = 4$, CNOT tree size $\gamma = 2$ and, $i \in \{0, 1, 2, 3\}$. For an address $|a_0 a_1 a_2 a_3\rangle$, the status registers of routers are set to $|\mathbf{L}_0\rangle = |a_0\rangle$, $|\mathbf{L}_1\rangle = |a_1\rangle$, $|\mathbf{X}_0\rangle = |\mathbf{X}_0'\rangle = |a_2\rangle$, and $|\mathbf{X}_1'\rangle = |a_3\rangle$. The colored line highlights the query branch for $x_0$. The number of repetitions in Stage II is $N/\lambda = 4$.

name suggests, the linear routers are at the top of the design. The middle of the layouts contains the CSWAP routers $\mathbf{X}_0$ and $\mathbf{X}_0'$ respectively. In Fig. 14a, the routers at the sides in the bottom of the figure correspond to the CNOT trees. By contrast, note that in Fig. 14b, the same qubits can be reused for the CSWAP routers $\mathbf{X}_1'$ and $\mathbf{X}_2'$. In comparison to Fig. 11b, clearly these layouts use far fewer qubits.

Now, we can analyze the T count, qubit count, and query infidelity for our framework for the planar layouts described here.

**Theorem** (Restatement of Theorem III.1). *Consider the quantum data lookup structure with the high-level scheme in Fig. 8 with $N$ memory locations. Let $n = \log N$, $\lambda = 2^{n-d}$ be the partition size and $\gamma = 2^{n-d-d'}$ be the size of a CNOT tree with $d' \le d \le n$. The infidelity of this circuit is*

$$O\left(\varepsilon_L \left(\frac{\gamma N}{\lambda} + \frac{N}{\lambda} \log \frac{\lambda}{\gamma}\right) + \varepsilon_s \log \frac{\lambda^2}{\gamma}\right.$$
$$+\varepsilon_I \left(\frac{N}{\lambda} \log N \left(\log \frac{N}{\gamma} + \gamma + \log \frac{\lambda}{\gamma}\right) + \text{polylog } \lambda\right)$$
$$+\varepsilon_c \frac{\gamma N}{\lambda} + \varepsilon_{cc} \frac{N}{\lambda} \log \frac{N}{\lambda}$$
$$\left.+\varepsilon_{cs} \left(\frac{N}{\lambda} \log \frac{\lambda}{\gamma} + \log^2 \frac{\lambda}{\gamma} + \log^2 \lambda\right)\right).$$
$$\tag{11}$$

*Moreover, the T count for this design is $O(\frac{N}{\gamma} + \frac{N}{\lambda} \log \frac{N}{\lambda} +$*

$\lambda)$, *and its qubit count is $O(\log \frac{N}{\lambda} + \lambda)$.*

*Proof.* First, consider Stage I where the linear routers do not contribute to the infidelity as their status is directly set by the address qubits. The CSWAP tree functions like a depth-$d'$ noise-resilient bucket-brigade QRAM. For the planar layout, the infidelity for the bucket-brigade QRAM is computed in Theorem IV.4. Using this, the depth-$d'$ CSWAP tree has infidelity $O(\text{poly}(d')\varepsilon_I + d'\varepsilon_L + d'\varepsilon_s + d'^2\varepsilon_{cs})$ in Stage I.

For the $\frac{N}{\lambda} = 2^d$ repetitions performed in Stage II, the infidelity comes both from gate errors when operations are performed as well as idling error on qubits that don't have gates on them. The $O(d)$ Toffoli gates used to implement the linear routers as per [4] each contribute $\varepsilon_{cc}$ to the infidelity. The error containment property of the CSWAP routers discussed in Section IV B implies that their contribution is only $d'\varepsilon_L$ for the long-range operations performed. For the CNOT tree with $\gamma$ leaves, its contribution amounts to $\varepsilon_L + \varepsilon_c$ for each node in the tree. Then, the overall gate error infidelity is

$$O\left(2^d \left(\varepsilon_{cc}\, d + \varepsilon_{cs}\, d' + \varepsilon_L \left(d' + 2^{n-d-d'}\right) + \varepsilon_c\, 2^{n-d-d'}\right)\right).$$
$$\tag{12}$$

The infidelity from the idling error is

$$O\left(2^d \left(\varepsilon_I n \left(d + d' + 2^{n-d-d'}\right)\right)\right), \tag{13}$$

where the terms correspond to the qubits that are not reset between repetitions. These are the router status

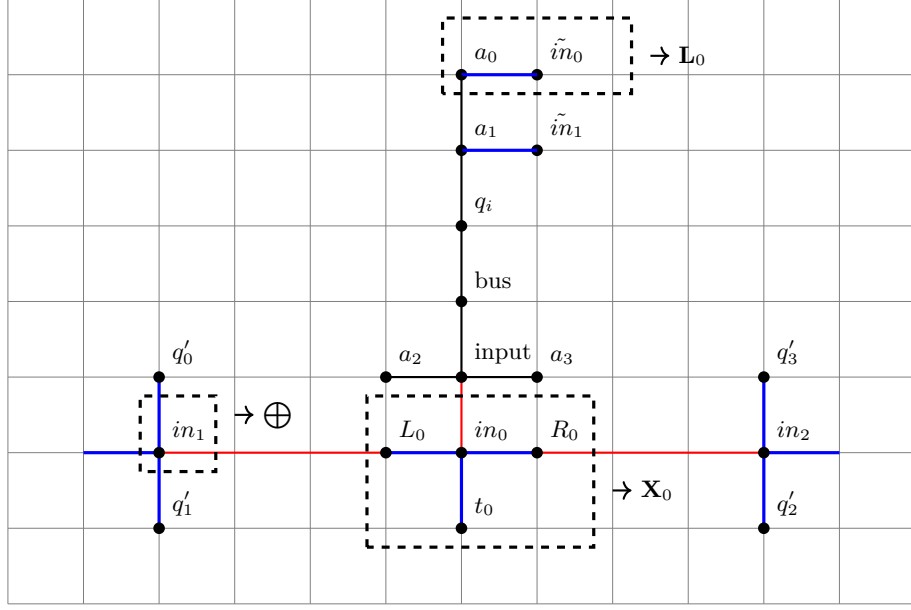

(a)

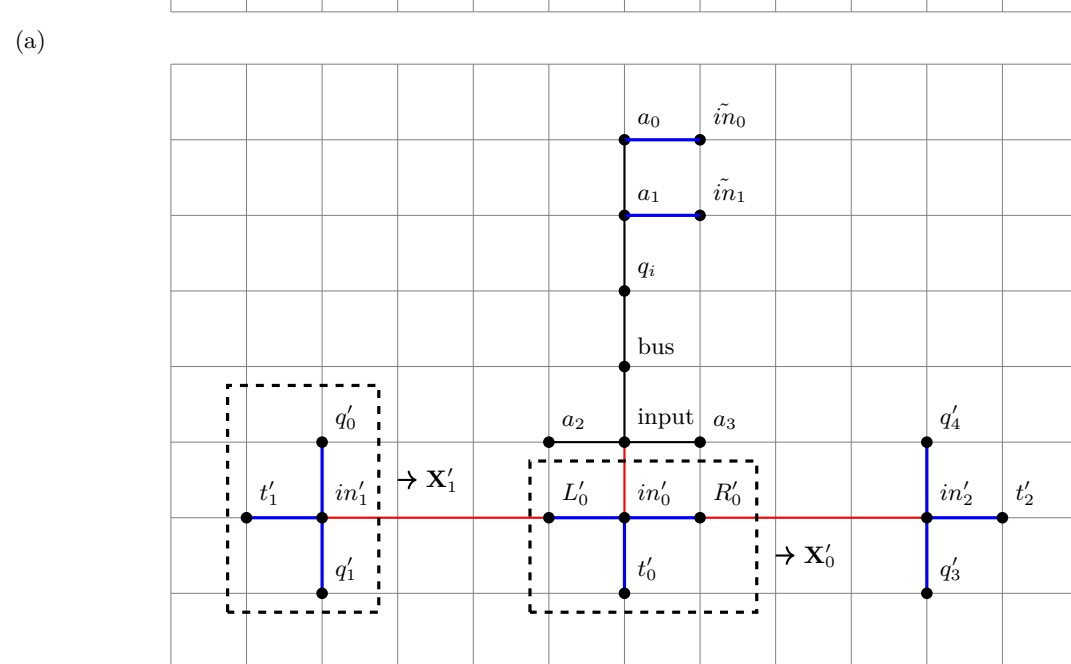

(b)

FIG. 14: (a) Qubit layout for the memory querying stage (II) of the unified quantum lookup architecture framework described in Fig. 8, Section III. (b) Qubit layout for the memory querying stage (III) of the unified quantum lookup architecture framework described in Fig. 8, Section III.

qubits for the linear routers and CSWAP routers along the query path, as well as the $\gamma$ intermediate registers $q_i'$ at the leaves of the activated CNOT tree.

Lastly, Stage III acts like a depth-$(n-d)$ bucket-brigade QRAM and applying Theorem IV.4 again, we obtain its infidelity as $O(\text{poly}(n-d)\varepsilon_I + (n-d)\varepsilon_L + (n-d)\varepsilon_s + (n-d)^2\varepsilon_{cs})$. Combining the infidelity from all stages, and replacing $d, d'$ and $n$ by $N, \lambda$, and $\gamma$ yields Eq. (11).

The overall T count for the design is

$$O\left(2^d(2^{d'} + d) + 2^{n-d}\right),\tag{14}$$

where the first term corresponds to the $2^d$ repetitions of Stage II each of which uses the depth-$d'$ CSWAP tree, and the last term comes from the depth-$(n-d)$ CSWAP tree in Stage III. Note that stage I has T count $O(2^{d'})$ and it is always asymptotically smaller than the Stage

II T count. Replacing $d, d'$ and $n$ yields a T count of $O(\frac{N}{\gamma} + \frac{N}{\lambda} \log \frac{N}{\lambda} + \lambda)$.

The overall qubit count for the design is

$$O(d + 2^{n-d}), \tag{15}$$

where the first term comes from the linear routers. The depth-$d'$ CSWAP tree and the $\gamma$ sized CNOT trees in total contain $O(2^{n-d})$ qubits. In Stage III, the qubits used for the CNOT trees are reused for the CSWAP routers accounted for in Stage II. Replacing $n$ and $d$ with $N$ and $\lambda$ yields an overall $O(\log \frac{N}{\lambda} + \lambda)$ qubit count. $\qquad\square$

## V. LARGE WORD SIZE

While all previous circuits considered reading a single bit of information from memory, in most real-world scenarios, one would want to retrieve multiple bits of classical information. In this section, we discuss two ways our general framework can be modified to handle the readout of multiple bits of information – (i) in parallel; and (ii) in sequence – while still maintaining a sub-linear scaling for memory size $N$. While the former has lower query infidelity, the latter has a lower T count. Determining where there exists a single multi-bit readout scheme that simultaneously minimizes T-count and infidelity or whether the need for two schemes remains a fundamental limitation of our framework is left for future work.

Throughout this section, we assume that the *word size*, i..e, number of bits to be read out from each memory location is $b$ and for ease of description, the designs will be described assuming $N = 16, \lambda = 4$ and $\gamma = 2$.

### A. Parallel multi-bit readout

To read $b$-bit words with parallel readout, we make $b$ copies of parts of our general framework from Fig. 13. Specifically, we create $b$ copies of the framework involving the register with $|q_i\rangle_q$, the CSWAP routers $\mathbf{X}_0$, $\mathbf{X}'_0$, $\mathbf{X}'_1$ and $\mathbf{X}'_2$ and the CNOT trees. Note that the three stages for data lookup proceed as described in Section III except that the $i$th copy will be used to access the $i$th bit of data, and at the end of Stage III all the $b$-bits will have been simultaneously retrieved. For $b = 2$, this modification is depicted in Fig. 15 where $|q_i\rangle_q$ is copied to the registers $|q_i^{(0)}\rangle_{q^{(0)}}$ and $|q_i^{(1)}\rangle_{q^{(1)}}$. A very high-level view of this schematic for $b = 6$ is given in Fig. 16 where each square labeled by $q_i^j$ contains the $j$th copy of the single-bit readout framework. The asymptotic scaling for infidelity, qubit, and T counts for parallel multi-bit readout is given below.

**Theorem V.1.** *Consider the quantum data lookup structure with the high-level scheme in Fig. 15 with $N$ memory locations. Let $b = 2^{d''}$ be the word size, $n = \log N$, $\lambda = 2^{n-d}$ be the partition size and $\gamma = 2^{n-d-d'}$ be the*

*size of a CNOT tree with $d' \le d \le n$. Also, let $\mathbf{I}$ be the infidelity of the single-bit readout framework from Theorem III.1. Then, the infidelity of this circuit is $O(b\mathbf{I})$. Moreover, the T count is $O(\log \frac{N}{\lambda} \frac{N}{\lambda} + b\frac{N}{\gamma} + b\lambda)$, and qubit count is $O(\log \frac{N}{\lambda} + b\lambda)$.*

*Proof.* The multi-bit parallel readout scheme amounts to containing a single copy of the linear routers followed by $b$ copies of the remainder of the single-bit readout framework. While the former uses $O(d)$ qubits, the latter uses $O(b2^{n-d})$ qubits. Additionally, as depicted in Fig. 16, each of $|q_i^j\rangle$ registers can be laid out such that they are separated by $O(2^{\frac{n-d}{2}})$ qubits in the planar grid. Hence, the overall qubit count is $O(d+b2^{n-d}+b2^{\frac{n-d}{2}}) = O(d + b2^{n-d}) = O(\log \frac{N}{\lambda} + b\lambda)$.

Similarly, the T count for this design is $O(2^d(d+b2^{d'}) + b2^{n-d}) = O(\log \frac{N}{\lambda} \frac{N}{\lambda} + b\frac{N}{\gamma} + b\lambda)$.

To analyze the infidelity for this design, let $\mathbf{I}_s$ represent the infidelity at stage $s$ as described in Theorem III.1. In Stage I, the infidelity is $O(b\mathbf{I}_I + \epsilon_L b2^{\frac{n-d}{2}}d')$, where the second term emerges from the fact that the $d'$ address bits used to set the CSWAP routers have to be copied $b$ times using long-range operations and routed through each corresponding copy of the single-bit readout framework as depicted in Fig. 15. Moving to Stage II, the infidelity is $O(b\mathbf{I}_{II} + 2^d(\epsilon_L b2^{\frac{n-d}{2}}))$, where the second term is for the $2^d$ times when $|q_i\rangle$ is transmitted to all the $|q_i^j\rangle$ registers. For Stage III, the infidelity is $O(b\mathbf{I}_{III}$ as each of the $b$ bits is retrieved from the corresponding copy of the readout framework when the procedure is finished. Aggregating the infidelities across these stages, observing that the extra terms only amount to an additional $(b\mathbf{I}_I)$ factor and following Eq. (11), we conclude that the total infidelity for parallel $b$-bit readout is $O(b\mathbf{I})$. $\qquad\square$

### B. Sequential multi-bit readout

To read $b$-bit words in sequence, we extend the single-bit framework in Fig. 13 by assuming that each memory location points to a $b$-qubit register instead of a single qubit one and repeating the readout procedure (i.e., Stage III from Section III) for each bit of data. For $b = 2$, this modification is presented in Fig. 17 where the CNOT tree now has depth $\log \frac{\lambda}{\gamma} + \log b$ and each leaf of the CNOT tree is connected to one of the $b$ bits of data. This allows the signal for a memory location to be accessed to be shared amongst the $b$ qubits. To achieve a compact design, the planar H-tree design is used to lay out the $b$ qubits of each memory location. A high-level view of this schematic is given in Fig. 18. To retrieve the $k$th bit of data, $|q_j'^{(k)}\rangle$ is swapped into the $q_j'$ location at iteration $k$ after which Stage III is applied to retrieve the data from the $q_j'$ register. The asymptotic scaling for infidelity, T count and qubit count for this design is given below.

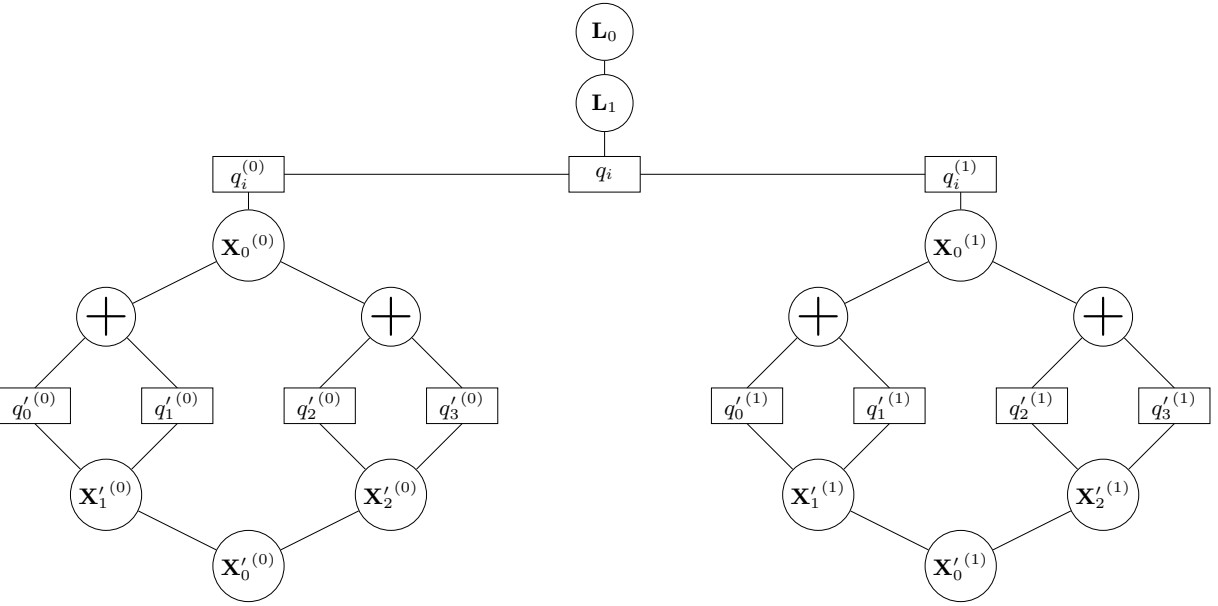

FIG. 15: A high-level scheme for parallel readout of word size $b = 2$, an extension to the scheme in Fig. 8. To avoid clutter, we label only the qubit variables and omit explicit register labels.

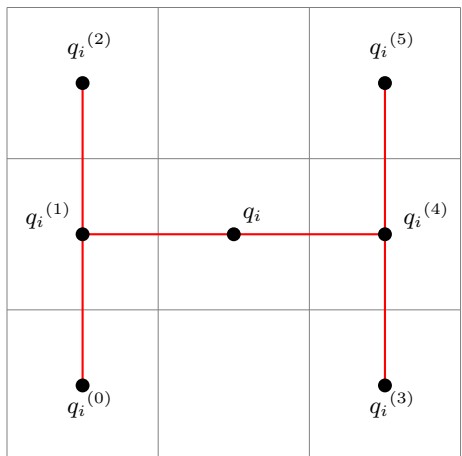

FIG. 16: A planar layout of our general quantum data lookup framework for parallel multi-bit readout with a word size of $b = 6$. Each square, labeled by $q_i^j$, encompasses the planar layout for each copy of the single-bit readout framework as depicted in Fig. 15. The red line indicates the path used for long-range operations responsible for diffusing the signal $q_i$ to all the $q_i^j$ instances.

**Theorem V.2.** *Consider the quantum data lookup structure with the high-level scheme in Fig. 17 with $N$ memory locations. Let $b = 2^{d''}$ be the word size, $n = \log N$. Let $b = 2^{d''}$ be the word size, $n = \log N$, $\lambda = 2^{n-d}$ be the partition size and $\gamma = 2^{n-d-d'}$ be the size of a CNOT tree with $d' \leq d \leq n$. Also, let $\mathbf{I}$ be the infidelity of the single-bit readout framework in Theorem III.1. Then, the infidelity of this circuit is $\tilde{O}(b\mathbf{I} + b^2\varepsilon_I)$. Moreover,*

*the $T$ count is $O(\frac{N}{\gamma} + \frac{N}{\lambda}\log\frac{N}{\lambda} + b\lambda)$, and qubit count is $O(\log\frac{N}{\lambda} + b\lambda)$.*

*Proof.* Before reading out the $b$ memory bits, the analysis follows the same as in Theorem III.1 which yields $O(b\mathbf{I})$ infidelity. There is an additional qubit idling error to be considered. It takes $O(2^{d''} + n - d + d'')$ time to transfer $b$ memory bits out sequentially. The idling qubits are the memory qubits and the CSWAP router status bits, hence the total idling error is $O(\varepsilon_I(2^{d''} + d')(2^{d''} + n - d + d'') = \tilde{O}(b^2\varepsilon_I)$.

The number of CSWAP routers and linear routers is unchanged for Stages I and II but the CSWAP routers are used $b$ times in Stage III. Hence, the T count is $O(\frac{N}{\gamma} + \frac{N}{\lambda}\log\frac{N}{\lambda} + b\lambda)$ where each term matches the count for each Stage respectively. The qubit count is $O(d + 2^{n-d+d''}) = O(\log\frac{N}{\lambda} + b\lambda)$ as the size of the H-tree design for Stages II and III are scaled by $b$ due to the deeper CNOT trees. $\square$

## VI. CONCLUSION

In this work, we construct a general quantum data lookup framework and describe how it can be arranged on a planar grid with local connectivity. Further, we demonstrate that for specific choices of parameters, this framework can be made to be simultaneously noise-resilient as well as resource-efficient i.e., having a sub-linear dependency on memory size for qubit count, T count, and query infidelity. The versatility of our framework in highlighted as it provides the blueprint for describing a family of circuits for quantum data lookup with various space, time, and noise resiliency trade-offs.

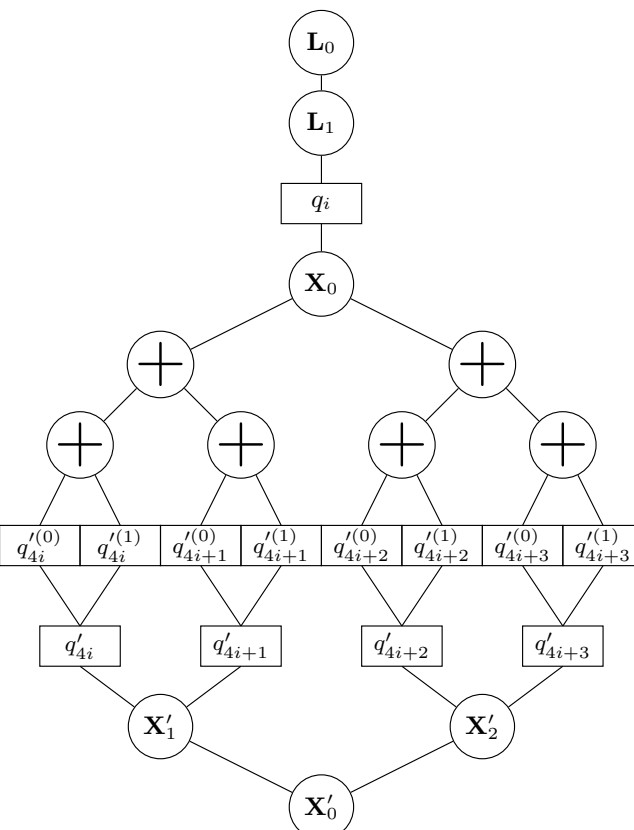

FIG. 17: A high-level scheme for sequential readout of word size $b = 2$, an extension to the scheme in Fig. 8.

resource-efficient than prior work irrespective of the constraints imposed by connectivity. For instance, assuming only nearest neighbor connectivity is among the more restrictive regimes and having some long-range connectivity for free improves query infidelity as we show in Appendix A. A thorough analysis of how our framework would perform for other qubit topologies is left for future work.

Our fine-grained analysis of noise resiliency can be put to use in practice by replacing those error terms with the noise models or behavior exhibited by the hardware of choice. Explicitly separating the different sources of error can aid in understanding how hardware behavior contributes to the overall query infidelity. This, in turn, can help identify any bottlenecks or design improvements that may be needed to improve hardware capabilities and implement quantum data lookup successfully.

Our framework recovers previous proposals for quantum data lookup for different choices of parameters. For example, with $\lambda = N$ and $\gamma = 1$, we recover the noise-resilient bucket brigade QRAM design; with $\lambda = \sqrt{N}$ and $\gamma = 1$, we recover a variant of the SELECT-SWAP design; and with $\lambda = 1$ and $\gamma = 1$ we recover the QROM design. Hence, we feel justified in considering our design to be a unifying framework. On the other hand, we acknowledge that one parameter regime that is not covered by our framework is the indicator function design from [23] that has $\sqrt{N}$ T-count and logarithmic query depth. We conjecture that this design has linear infidelity scaling but studying whether it can be made noise resilient while still maintaining the same T-count is left for future work.

Another interesting direction for future work is exploring the planar layout in a surface code, with different code distances for each layer of the CSWAP routers. Furthermore, since the CNOT tree only propagates $Z$ errors, one could consider using an unbalanced surface code that produces $X$ errors with a higher probability.

While we presented a straightforward method to reset the qubits used for table lookup, it will result in a doubling of the T count and query infidelity. Another approach would be to use measurement-based uncomputation as discussed in [21, Appendix C]. This will lead to an increase in classical processing to deal with the mid-circuit measurement outcomes but not add too much overhead in terms of unitary operations. The further study needed to determine whether this will maintain the sub-linear scaling in infidelity is left as an open question.

There exists an implicit assumption that the planar layout and local connectivity versions of both our framework and the bucket-brigade QRAM require the use of error correction to drive gate errors below some critical threshold. Without this, the designs can fail to be error-resilient. The bottleneck here is the difficulty in implementing the entanglement distillation that is used for long-range operations. For distillation to be effective, it is necessary that the initial error of noisy Bell pairs – determined solely by gate errors – be below some thresh-

As the community moves beyond NISQ architectures to designing error-corrected implementations of quantum hardware [41–45] there is an urgent need to design highly resource-optimal components for quantum applications. For instance, initially, T gates would be considered an expensive resource so the focus would be on minimizing the T count. However, future improvements in large-scale quantum hardware and quantum error correction could dramatically reduce the cost of T gates, which would shift the focus to minimizing the query infidelity. Having sub-linear infidelity scaling in this setting would imply that smaller distance error correcting codes can be used to minimize overall errors and lead a to reduced qubit count. In this way, we expect the space-time trade-offs that result from this work to be repeatedly harnessed for the realization of end-to-end implementations of quantum algorithms using the evolving state-of-the-art hardware of its time [46].

The circuit designs presented in this work consider limited local connectivity with qubits laid out in a planar grid. To the best of our knowledge, there has been no prior comprehensive investigation into a resilient quantum lookup table framework for real-world applications that takes into account both resource limitations and local connectivity. However, we believe that our general framework would be noise resilient and can be more

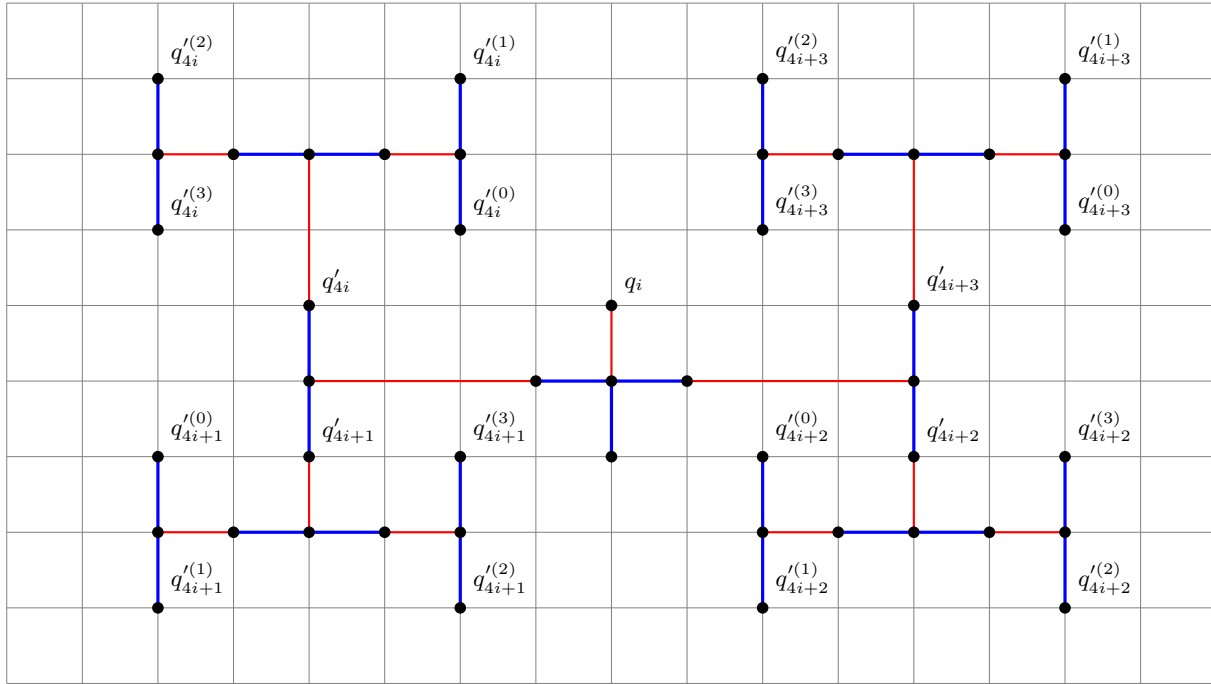

FIG. 18: A planar layout of the multi-bit register for each memory location in the quantum data lookup framework with sequential readout having a word size of $b = 4$.

old. We leave as an open question whether this threshold can be raised by more sophisticated techniques such as using quantum repeaters negating the need for error correction.

## ACKNOWLEDGMENTS

The authors thank the anonymous referee for their detailed and constructive suggestions. SZ conducted the research during his internship at Microsoft. Additionally, SZ is supported by the National Science Foundation CAREER award (grant CCF-1845125), and part of the editing work was performed while SZ was visiting the Institute for Pure and Applied Mathematics (IPAM), which is supported by the National Science Foundation (Grant No. DMS-1925919). SZ also acknowledges support from the U.S. Department of Energy, Office of Science, Accelerated Research in Quantum Computing Centers, Quantum Utility through Advanced Computational Quantum Algorithms, grant No. DE-SC0025572.

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

arXiv:2211.15465 [quant-ph].

## Appendix A: Limited long-range connections

In this section, we consider variations in qubit architectures that allow for limited long-range connectivity. Examples include superconducting qubits with long-range connections for LDPC codes or those where qubits can be coupled with photons that can be used to create high-quality Bell states. We study how the assumption of allowing only a few gates to be able to act long-range without overhead impacts infidelity. Unlike in the main text, we assume that there are only two ways in which to perform long-range interactions: (i) using a limited number of almost perfect Bell pairs that can be used to perfectly perform long-range operations; and (ii) creating noisy GHZ states linked between source and target qubits as described in Section III.

### 1. Bucket-brigade model

First, consider the planar bucket-brigade model from Section IV A. We allow the first $k$ level of CSWAP routers the ability to perform long-range operations without any errors and produce a modified version of Theorem IV.4.

**Corollary A.1.** *Let the first $k$ level routers in the bucket-brigade QRAM of Fig. 11a have the ability to perform long-range operations without any overhead. For $N$ memory locations, the improved fine-grained infidelity of the basic planar layout (Fig. 11b) scales as*

$$O\left(2^{-\frac{k}{2}}\sqrt{N}\varepsilon_Q \log N + \varepsilon_s \log N + \varepsilon_{cs} \log^2 N + \varepsilon_I \log^2 N\right). \tag{A1}$$

*Proof.* The result largely follows from the proof of Theorem IV.4 along with two modifications. First, adjust the initial value of $\ell$ in Eq. (7) from 1 to $k$ as all levels up to $k$ will not contribute to query infidelity. Additionally, as the remaining long-range operations only use GHZ states as a resource, its contribution to infidelity is $m \cdot \varepsilon_Q$ for GHZ states of length $m$ from Lemma IV.1. As $m < \sqrt{N}$, we can upper bound this contribution as $O(\sqrt{N}\varepsilon_Q)$. Since no magic state distillation is applied, the total idling time for each CSWAP router $R_\ell$ is $O(T - \ell)$, which leads to an overall $\log^2 N$ error contribution from the idling error. This gives the desired result. ∎

When $k$ equals $\log N$, the planar layout achieves the expected $O(\log^2 N)$ infidelity scaling from [24].

### 2. General framework

We study the effect of allowing the first $k$ level of CSWAP + CNOT routers in Stages I and II of the general framework from Section III the ability to perform long-range operations without any error and produce a modified version of version of Theorem III.1.

**Corollary A.2.** *Consider the quantum data lookup structure with the high-level scheme in Fig. 8 with $N$ memory locations. Let $n = \log N$, $\lambda = 2^{n-d}$ be the partitions size, and $\gamma = 2^{n-d-d'}$ be the size of a CNOT tree with $d' \le d \le n$. Let the first $k$-levels of the CSWAP + CNOT routers in Stages I and II have the ability to perform long-distance SWAP gates without any overhead. Let $\mathcal{E}$ denote the sum of all but $\varepsilon_L$ error in Theorem III.1, then the infidelity of the circuit is*

$$\tilde{O}\left(\varepsilon_Q\left(2^{-\frac{k}{2}}\sqrt{\lambda} + 2^{-\frac{k}{2}}\frac{N}{\sqrt{\lambda}} + \frac{\gamma N}{\lambda}\right) + \mathcal{E}\right), \tag{A2}$$

*for $k \le d'$, and*

$$\tilde{O}\left(\varepsilon_Q\left(2^{-k}N + 2^{-\frac{k}{2}}\sqrt{\lambda}\right) + \mathcal{E}\right), \tag{A3}$$

*for $d' < k \le n - d$.*

*Proof.* For $k \leq d'$, in stage I the GHZ error induced from qubit error is $d'2^{\frac{n-d-k}{2}}\varepsilon_Q$, and in Stage II the GHZ error is $2^d(2^{\frac{n-d-k}{2}} + 2^{n-d-d'})\varepsilon_Q$. The remaining error analysis follows from Theorem III.1, and combining these yields the desired GHZ error.

For $k > d'$, in stage I there is no GHZ error, in stage II the GHZ error comes from the CNOT tree, which becomes $2^d \cdot 2^{n-d-d'-(k-d')}\varepsilon_Q$. The remaining error analysis follows from Theorem III.1, and combining these yields the desired GHZ error. $\qquad\square$

Corollary A.2 illustrates the challenging nature of achieving optimal scaling in various aspects by balancing the parameters $\lambda$ and $\gamma$ with a budget of performing $O(2^k)$ long-range operations without overhead. It becomes apparent that striking the perfect balance is a formidable task as it adds parameter that needs to be optimized.

However, for some limited flexibility in choices for $d = \log\left(\frac{N}{\lambda}\right)$, $d' = \log\left(\frac{\lambda}{\gamma}\right)$, and the long-range budget $2^k$, we try to provide some additional insights. For instance, within the realm of $\tilde{O}(N^{3/4})$ T count, we tabulate how the exponent determining how infidelity scales for increasing values of $k \in \{0, d'/4, d'/2, 3d'/4, d\}$ in Tables V to IX respectively. Specifically for the green regions in these tables, we find that:

1. When $d'$ is relatively small, specifically when $d' \leq n/4$, the presence of long-range connectivity does not decrease infidelity. This is because the primary source of error originates from the CNOT tree section of the circuit.

2. Any further decrease in infidelity becomes unattainable when $k$ exceeds $d'/2$. This is because, when $k > d'/2$, the predominant factor contributing to the error is the idling error by Eq. (A2).

3. The greater the value of $d'$ that can be accommodated, the more favorable the infidelity scaling becomes. This is because larger values of $d'$ serve to diminish both the long-range errors as well as the idling error for the CNOT tree.

4. Increasing the value of $d$ leads to a more favorable qubit count, as the qubit count scales as $O(2^{n-d})$. However, this sets up a trade-off between selecting a larger value of $d$ for improved qubit count or a larger value of $d'$ for reduced infidelity within the $O(N^{3/4})$ T-count regime.

Extending the analysis naturally allows the initial $k$ levels in the route-in procedure to perform long-range operations without overhead, while permitting the final $k'$ levels in the route-out procedure to do the same. In a practical context, this can be likened to having a long-range budget of $2^{d+k}$ for the route-in and a long-range budget of $2^{k'}$ for the route-out process. The objective is to strike a balance between the values of $k$ and $k'$ to attain the most favorable scaling of infidelity.

We can briefly argue that it is not useful to separately consider $k$ vs. $k'$ in the regime that yields sublinear infidelity and T count. This is because the long-range error from route-out only dominates that of route-in when $k > 2d$, at which point a non-zero $k'$ is required to suppress the long-range error from the route-out procedure. However, we observed that $k$ is only meaningful for $k \leq \frac{d'}{2}$, and it only improves the infidelity for cases where $d \leq \frac{d'}{2}$, which contradicts the condition where $k > 2d$.

| d \ d' | 0 | 1/8 | 1/4 | 3/8 | 1/2 | 5/8 | 3/4 | 7/8 | 1 |
|---|---|---|---|---|---|---|---|---|---|
| 0 | 1 | 7/8 | 3/4 | 5/8 | 1/2 | 1/2 | 1/2 | 1/2 | 1/2 |
| 1/8 | 1 | 7/8 | 3/4 | 5/8 | 9/16 | 9/16 | 9/16 | 9/16 | |
| 1/4 | 1 | 7/8 | 3/4 | 5/8 | 5/8 | 5/8 | 5/8 | | |
| 3/8 | 1 | 7/8 | 3/4 | 11/16 | 11/16 | 11/16 | | | |
| 1/2 | 1 | 7/8 | 3/4 | 3/4 | 3/4 | | | | |
| 5/8 | 1 | 7/8 | 13/16 | 13/16 | | | | | |
| 3/4 | 1 | 7/8 | 7/8 | | | | | | |
| 1/8 | 1 | 15/16 | | | | | | | |
| 1 | 1 | | | | | | | | |

| T count | 1/2 | 5/8 | 3/4 | 7/8 | 1 |
|---|---|---|---|---|---|

TABLE V: The exponent of infidelity scaling is examined for $N = 2^n$ with zero long-range budget, while varying $d$ and $d'$ subject to the constraint $d + d' \leq n$, where $\lambda = 2^{n-d}$ and $\gamma = 2^{n-d-d'}$.

| d\d' | 0 | 1/8 | 1/4 | 3/8 | 1/2 | 5/8 | 3/4 | 7/8 | 1 |
|---|---|---|---|---|---|---|---|---|---|
| 0 | 1 | 7/8 | 3/4 | 5/8 | 1/2 | 27/64 | 13/32 | 25/64 | 3/8 |
| 1/8 | 1 | 7/8 | 3/4 | 5/8 | 1/2 | 31/64 | 15/32 | 29/64 | |
| 1/4 | 1 | 7/8 | 3/4 | 5/8 | 9/16 | 35/64 | 17/32 | | |
| 3/8 | 1 | 7/8 | 3/4 | 41/64 | 5/8 | 39/64 | | | |
| 1/2 | 1 | 7/8 | 3/4 | 45/64 | 11/16 | | | | |
| 5/8 | 1 | 7/8 | 25/32 | 49/64 | | | | | |
| 3/4 | 1 | 7/8 | 27/32 | | | | | | |
| 1/8 | 1 | 59/64 | | | | | | | |
| 1 | 1 | | | | | | | | |

TABLE VI: The exponent of infidelity scaling is examined for $N = 2^n$ with $k = d'/4$, while varying $d$ and $d'$ subject to the constraint $d + d' \leq n$.

| d\d' | 0 | 1/8 | 1/4 | 3/8 | 1/2 | 5/8 | 3/4 | 7/8 | 1 |
|---|---|---|---|---|---|---|---|---|---|
| 0 | 1 | 7/8 | 3/4 | 5/8 | 1/2 | 3/8 | 5/16 | 9/32 | 1/4 |
| 1/8 | 1 | 7/8 | 3/4 | 5/8 | 1/2 | 13/32 | 3/8 | 11/32 | |
| 1/4 | 1 | 7/8 | 3/4 | 5/8 | 1/2 | 15/32 | 7/16 | | |
| 3/8 | 1 | 7/8 | 3/4 | 5/8 | 9/16 | 17/32 | | | |
| 1/2 | 1 | 7/8 | 3/4 | 21/32 | 5/8 | | | | |
| 5/8 | 1 | 7/8 | 3/4 | 23/32 | | | | | |
| 3/4 | 1 | 7/8 | 13/16 | | | | | | |
| 1/8 | 1 | 29/32 | | | | | | | |
| 1 | 1 | | | | | | | | |

TABLE VII: The exponent of infidelity scaling is examined for $N = 2^n$ with $k = d'/2$, while varying $d$ and $d'$ subject to the constraint $d + d' \leq n$.

| d\d' | 0 | 1/8 | 1/4 | 3/8 | 1/2 | 5/8 | 3/4 | 7/8 | 1 |
|---|---|---|---|---|---|---|---|---|---|
| 0 | 1 | 7/8 | 3/4 | 5/8 | 1/2 | 3/8 | 1/4 | 11/64 | 1/8 |
| 1/8 | 1 | 7/8 | 3/4 | 5/8 | 1/2 | 3/8 | 9/32 | 15/64 | |
| 1/4 | 1 | 7/8 | 3/4 | 5/8 | 1/2 | 25/64 | 11/32 | | |
| 3/8 | 1 | 7/8 | 3/4 | 5/8 | 1/2 | 29/64 | | | |
| 1/2 | 1 | 7/8 | 3/4 | 5/8 | 9/16 | | | | |
| 5/8 | 1 | 7/8 | 3/4 | 43/64 | | | | | |
| 3/4 | 1 | 7/8 | 25/32 | | | | | | |
| 1/8 | 1 | 57/64 | | | | | | | |
| 1 | 1 | | | | | | | | |

TABLE VIII: The exponent of infidelity scaling is examined for $N = 2^n$ with $k = 3d'/4$, while varying $d$ and $d'$ subject to the constraint $d + d' \leq n$.

| d\d' | 0 | 1/8 | 1/4 | 3/8 | 1/2 | 5/8 | 3/4 | 7/8 | 1 |
|---|---|---|---|---|---|---|---|---|---|
| 0 | 1 | 7/8 | 3/4 | 5/8 | 1/2 | 3/8 | 1/4 | 1/8 | 0 |
| 1/8 | 1 | 7/8 | 3/4 | 5/8 | 1/2 | 3/8 | 1/4 | 1/8 | |
| 1/4 | 1 | 7/8 | 3/4 | 5/8 | 1/2 | 3/8 | 1/4 | | |
| 3/8 | 1 | 7/8 | 3/4 | 5/8 | 1/2 | 3/8 | | | |
| 1/2 | 1 | 7/8 | 3/4 | 5/8 | 1/2 | | | | |
| 5/8 | 1 | 7/8 | 3/4 | 5/8 | | | | | |
| 3/4 | 1 | 7/8 | 3/4 | | | | | | |
| 1/8 | 1 | 7/8 | | | | | | | |
| 1 | 1 | | | | | | | | |

TABLE IX: The exponent of infidelity scaling is examined for $N = 2^n$ with $k = d'$ long-range budget.