# OpenReview forum: "A unified architecture for quantum lookup tables"
_purdue.edu/Purdue_University/PQAI/2025/Symposium — PQAI 2025 Oral_

### Official Review · Reviewer_xsXA · 2025-07-22

**Rating:** 9
**Confidence:** 4

**Review:**

This paper explores the design of a quantum lookup table, a critical component that allows a quantum computer to access classical data and is the foundation of many quantum machine learning algorithms. The main contribution is a flexible and general architecture for a quantum lookup table. By adjusting the parameters, it can achieve optimal tradeoffs between qubit count, non-Clifford gates, and error resilience. This framework can incorporate all previous works. Additionally, it does not require all-to-all connectivity, only 2D local connectivity.

Overall, I believe this is a strong paper with non-trivial results and wide-ranging applications. Therefore, I recommend accepting it as an oral presentation.

---

### Official Review · Reviewer_GX6G · 2025-07-24
**This paper presents a unified parameterized architecture for quantum lookup tables that encompasses prior work while enabling novel resource tradeoffs between qubits, non-Clifford gates, and error resilience.**

**Rating:** 9
**Confidence:** 2

**Review:**

Strengths:

The paper provides a genuinely unified framework that recovers all major prior quantum lookup table architectures as special cases.

Achieves the notable result of simultaneous sublinear scaling in infidelity, T-gate count, and qubit count with only local connectivity

The local connectivity requirement is highly relevant for near-term quantum devices

Fine-grained error analysis separating different error sources provides actionable insights for hardware implementations

Comprehensive theoretical analysis with detailed proofs

Thorough comparison with existing approaches across multiple metrics

Weaknesses:

The entanglement distillation protocol for long-range operations adds significant overhead that may limit practical applicability.

Purely theoretical work with no experimental demonstration or simulation results.

Missing analysis of how the approach scales with realistic noise parameters from current quantum devices.

Limited discussion of how the approach compares to classical alternatives in practical scenarios.

---

### Decision · Program_Chairs · 2025-07-29

Accept (Oral)